# 3DMedAgent: Unified Perception-to-Understanding for 3D Medical Analysis

**Ziyue Wang** [1 2]  **Linghan Cai** [3]  **Chang Han Low** [1]  **Haofeng Liu** [1]  **Junde Wu** [4]  **Jinyu Wang** [2]  **Rui Wang** [2]
**Lei Song** [2]  **Jiang Bian** [2]  **Jingjing Fu** [2 *]  **Yueming Jin** [1 *]

## Abstract

3D CT analysis spans a continuum from low-level perception to high-level clinical understanding. Existing 3D-oriented analysis methods adopt either isolated task-specific modeling or task-agnostic end-to-end paradigms to produce one-hop outputs, impeding the systematic accumulation of perceptual evidence for downstream reasoning. In parallel, recent multimodal large language models (MLLMs) exhibit improved visual perception and can integrate visual and textual information effectively, yet their predominantly 2D-oriented designs fundamentally limit their ability to perceive and analysis volumetric medical data. To bridge this gap, we propose 3DMedAgent, an unified agent that enables 2D MLLMs to perform general 3D CT analysis without 3D-specific fine-tuning. 3DMedAgent coordinates heterogeneous visual and textual tools through a flexible MLLM agent, progressively decomposing complex 3D analysis into tractable sub-tasks that transition from global to regional views, from 3D volumes to informative 2D slices, and from visual evidence to structured textual representations. Central to this design, 3DMedAgent maintains a long-term structured memory that aggregates intermediate tool outputs and supports query-adaptive, evidence-driven multi-step reasoning. We further introduce the DeepChestVQA benchmark for evaluating unified perception-to-understanding capabilities in 3D thoracic imaging. Experiments across over 40 tasks demonstrate that 3DMedAgent consistently outperforms general, medical, and 3D-specific MLLMs, highlighting a scalable path toward general-purpose 3D clinical assistants. Code and data are available here.

[1]National University of Singapore [2]Microsoft Research [3]TUD Dresden University of Technology [4]University of Oxford. Correspondence to: Jingjing Fu <Jingjing.Fu@microsoft.com>, Yueming Jin <ymjin@nus.edu.sg>.

*Proceedings of the $43^{rd}$ International Conference on Machine Learning*, Seoul, South Korea. PMLR 306, 2026. Copyright 2026 by the author(s).

## 1. Introduction

3D medical imaging, especially computed tomography (CT) imaging, serves as a fundamental modality in modern medicine (Haydel et al., 2000). It provides volumetric views of anatomy, enabling a granular assessment of organ and tissue characteristics (Hasbun et al., 2001; Group, 2022). As shown in Figure 1, analysis within this domain spans a diverse range of tasks, from fundamental visual perception such as organ size measurement to higher-level clinical understanding such as tumor staging. Despite the clinical utility, the process requires experts to perform exhaustive slice-by-slice review through dense volumetric data, leading to a growing burden and potentially increasing the risk of diagnostic errors under heavy workload (Patel et al., 2020; Bruls & Kwee, 2020). Therefore, there is an urgent need for AI-assisted 3D imaging analysis to streamline time-consuming volumetric review and provide reliable decision support (Oren et al., 2020; Topol, 2019).

Prior works mainly address 3D image analysis tasks in isolation. On the one hand, segmentation and grounding models (Isensee et al., 2021; Wasserthal et al., 2023) have been developed to provide reliable anatomical localization, enabling low-level measurements and recognition. On the other hand, multi-modal approaches (Lin et al., 2024; Hu et al., 2025) facilitates visual and medical reasoning by learning joint embeddings of volumetric CT and clinical text. These aligned embeddings can then be leveraged for downstream tasks such as visual question answering (VQA) and report generation. Conversely, clinical decision-making is inherently sequential and interdependent: medical understanding builds on accurate perception, with reliable measurements and lesion recognition providing essential evidence for subsequent reasoning. For example, as illustrated in Figure 1, assessing fatty liver relies on precise measurements of organ size and attenuation, while tumor risk analysis depends on accurate recognition of lesion type and extent. This dependency motivates a unified solution that begins with precise perception with complementary evidence to enable comprehensive medical understanding.

Recently, multimodal large language models (MLLMs) (Liu et al., 2023; Team et al., 2023) have demonstrated strong capabilities for information integration and visual reasoning,

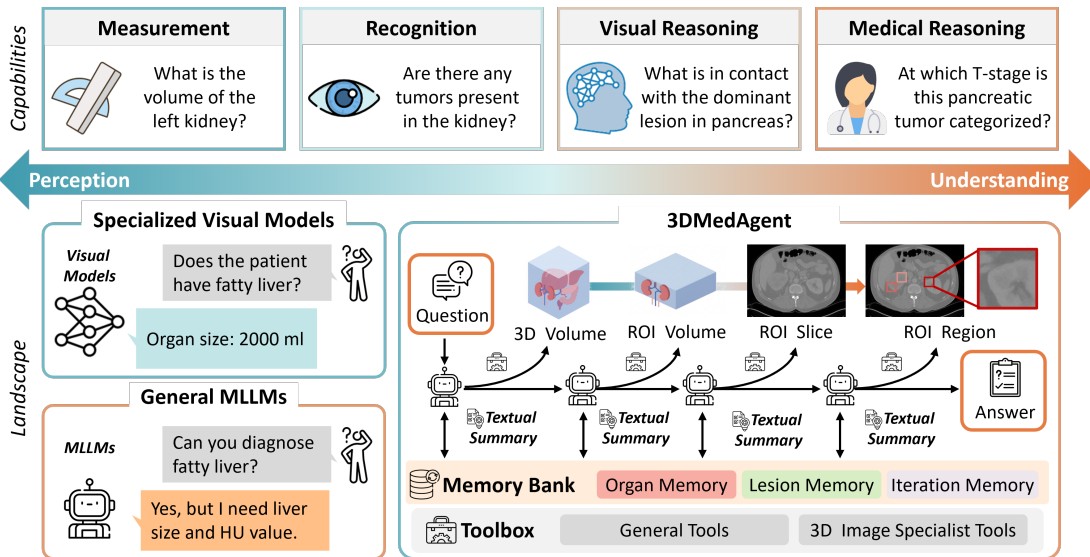

*Figure 1.* The illustration of existing tasks and methods in 3D medical image analysis. 3DMedAgent adaptively invokes visual tools for task-specific perception and analysis, and summarizes the outputs as compact evidence stored in shared memory for subsequent reasoning.

offering a promising direction toward unified 3D clinical assistants. However, direct applications of MLLMs to 3D image analysis remains challenging. Most MLLMs are designed for 2D inputs, while processing 3D volumes as image sequences is inefficient and discard spatial context essential for volumetric understanding. Some approaches adapt MLLMs with 3D vision encoders and perform 3D instruction tuning (Wu et al., 2025; Blankemeier et al., 2024). They compress huge 3D volumes into limited tokens for 2D MLLM backbones. However, this tokenization can blur fine-grained anatomy and encourage shortcut pattern-matching rather than genuine 3D understanding (Zech et al., 2018; Luo et al., 2025). With scarce and heterogeneous 3D data, these models are often brittle under clinical domain shift, making accurate measurement and recognition particularly difficult and further hindering reliable medical reasoning.

Tool-augmented agents enable MLLMs to invoke external tools and iteratively gather task-relevant evidence (Yao et al., 2022; Schick et al., 2023; Wang et al.). For 3D medical image analysis, this paradigm decouples perception from understanding, mitigating the 2D-oriented MLLMs' bottleneck in volumetric reasoning. Motivated by this, we present 3DMedAgent, a unified 3D medical imaging analysis agent that advances from perception to understanding without task-specific 3D MLLM training. 3DMedAgent follows a query-adaptive evidence-seeking loop. It invokes visual and textual tools as needed, narrows from global context to regional evidence, and converts 3D volumes into informative 2D slices. Each intermediate result is distilled into compact structured evidence and stored in a long-term shared memory to support multi-step reasoning. Concretely, 3DMedAgent performs Organ-Aware Memory Initialization (OAMI), Coarse-to-Fine Lesion Targeting (CFLT), and an on-demand Think-with-1-Slice Loop (T1S-Loop) to acquire, verify, and update evidence in memory. With this evolving memory, the agent refines the final answer with explicit supporting evidence. In summary, our contributions are:

- We propose 3DMedAgent, a unified solution that enables 2D MLLMs to perform general 3D CT analysis from perception to understanding without 3D-specific fine-tuning

- We introduce an evidence-centric long-term memory that distills heterogeneous tool outputs into compact textual evidence, enabling query-conditioned cue acquisition and aggregation for multi-step 3D reasoning.

- We also introduce a DeepChestVQA benchmark to enable broader evaluation. Experiments show that 3DMedAgent consistently outperforms general, medical, and 3D-specific MLLMs across 40+ 3D medical tasks, achieving an overall 20% accuracy gains.

## 2. Related Work

### 2.1. Medical Vision-Language Models

Early studies (Wang et al., 2018; Sharma et al., 2021) primarily perform feature fusion for medical images and clinical text, while subsequent works (Zhang et al., 2023a; Lin et al., 2023) adopt CLIP-style pretraining to obtain multimodal representations from large-scale biomedical corpora. With the rapid advancement of MLLMs (Liu et al., 2023; Alayrac et al., 2022), recent efforts have extended them to medical MLLMs that support a broad range of tasks, including visual question answering, report generation, and clinical reasoning (Tu et al., 2024; Li et al., 2023; Sellergren et al., 2025).

However, most existing medical MLLMs are designed for 2D imaging and do not natively support volumetric data.

Several works have explored extending MLLMs to 3D medical imaging: RadFM (Wu et al., 2025) proposes a unified encoder capable of processing both 2D and 3D scans, while Merlin (Blankemeier et al., 2024) and CT-CHAT (Hamamci et al., 2024b) incorporate 3D vision encoders into existing 2D MLLM frameworks. However, extending medical MLLMs to 3D imaging typically entails substantial computational cost and reliance on large-scale annotated volumetric data. Under such constraints, existing 3D-capable models, often trained with limited model capacity, tend to exhibit weaker instruction-following fidelity and general reasoning performance compared to their 2D counterparts, limiting their practical applicability in clinical settings.

### 2.2. Medical Agentic Systems

Medical agentic systems have been developed to address the limitations of single MLLM, and have demonstrated promising performance across a range of clinical scenarios like radiology (Fallahpour et al., 2025; Chen et al., 2024b), pathology (Ghezloo et al., 2025; Chen et al., 2025a) and surgery (Low et al., 2025a;b). These approaches typically either leverage multiple MLLM-based agents to conduct debate or majority voting (Tang et al., 2024; Kim et al., 2024; Xia et al., 2025), or leverage specialized medical tools to enable structured reasoning (Li et al., 2024; Wang et al., 2025; Zhu et al., 2025). However, extending such agentic systems to 3D medical imaging remains challenging, as it requires 2D MLLMs to effectively extract, integrate, and reason over information from volumetric data.

### 2.3. Medical Benchmarks

VQA is a widely used paradigm for evaluating multimodal medical models and systems. Early benchmarks such as VQA-RAD (Lau et al., 2018), SLAKE (Liu et al., 2021), and PathVQA (He et al., 2020) focus on 2D medical images with limited question types, mainly covering simple modality or organ-level classification. Subsequent datasets including PMC-VQA (Zhang et al., 2023b), OmniMedVQA (Hu et al., 2024), and MedXpertQA (Zuo et al., 2025) substantially scale up dataset size, but remain restricted to 2D images.

Datasets with paired 3D volumes and reports (Ji et al., 2022; Hamamci et al., 2024a; Zhang et al., 2024) enable the development of 3D VQA benchmarks. However, most existing 3D VQA datasets (Gai et al., 2025; Bai et al., 2024) lack hierarchical QA structures and explicit question-type categorization, hindering comprehensive evaluation of 3D understanding. Recently, DeepTumorVQA (Chen et al., 2025b) introduced a large-scale benchmark with 29 medical tasks of 4 dimensions, but mainly focuses on abdominal region with diseases largely limited to cysts and tumors. This limits evaluation across anatomical sites, leaving thoracic evaluation underexplored despite its great pathological diversity.

*Table 1.* Overview of the DEEPCHESTVQA benchmark.

| Dataset Overview | |
| --- | --- |
| CT scans | 922 |
| VQA pairs | 1020 |
| *Question Taxonomy* | | |

| Question type | #Subtypes | #VQA pairs |
| --- | --- | --- |
| Recognition | 3 | 180 |
| Visual Reasoning | 8 | 480 |
| Medical Reasoning | 6 | 360 |

## 3. DeepChestVQA Benchmark Construction

To complement abdominal-focused evaluations, we introduce **DeepChestVQA**, a comprehensive chest CT benchmark designed to complement abdominal-focused evaluations. Following the hierarchical structure of DeepTumorVQA, DeepChestVQA spans 17 capability dimensions across 1,020 VQA pairs with corresponding organ and lesion masks. Detailed statistics are provided in Table 1.

Specifically, raw CT volumes are sourced from CT-RATE. ReXGroundingCT (Baharoon et al., 2025) provides lesion masks for a subset of these scans. For this subset, we obtain organ masks using advanced segmentation models like VISTA3D (He et al., 2025) and TotalSegmentator (Wasserthal et al., 2023), and manually refine some low-quality masks. Following DeepTumorVQA, *recognition* and *visual-reasoning* questions are generated from organ and lesion masks. *Measurement* tasks are excluded in DeepChestVQA, as they rely directly on segmentation masks already used for question generation. For medical-reasoning questions, clinically meaningful chest CT problems requiring structured reasoning are curated; relevant indicators are computed from the masks to derive answers, followed by manual verification and calibration. For closed-set questions in the dataset, we ensure an equal number for each option and randomize the option ordering in multiple-choice questions (MCQs). More details are provided in Appendix A.

## 4. Methodology

3DMedAgent equips a 2D MLLM agent $A_\theta$ with three components to form a long-term shared memory that distills heterogeneous tool outputs into explicit evidence. This memory is iteratively updated and reused to support query-adaptive evidence seeking and selection for 3D volumetric analysis. First, we perform Organ-Aware Memory Initialization (OAMI) to initialize the MLLM agent with compact organ-level descriptions. For lesion-related queries, the agent then conducts Coarse-to-Fine Lesion Targeting (CFLT) to pro-

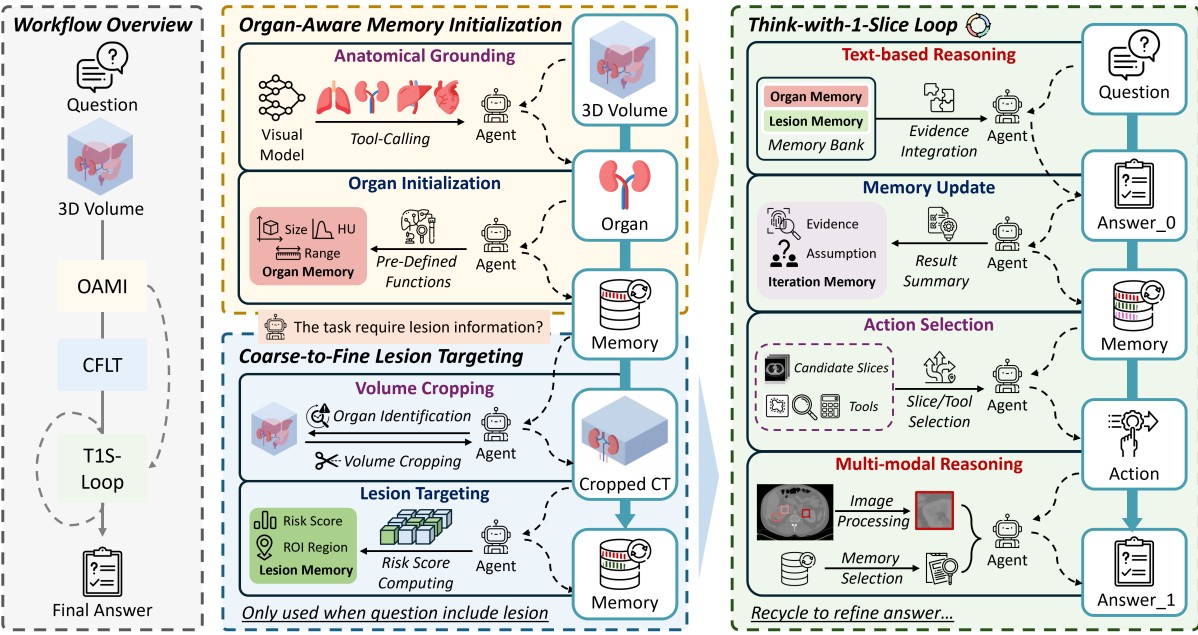

*Figure 2.* The overall framework of 3DMedAgent. A 2D MLLM agent iteratively interacts with heterogeneous tools, distills their outputs into structured evidence, and updates a shared memory for reliable 3D medical image analysis. Subtitle colors follow the same agentic action-type scheme, and colors denote agentic action types: tool use (purple), memory (blue), and reasoning (red).

gressively localize candidate regions of interests (ROIs) and target informative slices. Finally, 3DMedAgent conducts Think-with-1-Slice Loop (T1S-Loop) if necessary: at each iteration, the agent adaptively selects one slice, conduct slice-level reasoning to verify evidence and updates the memory to refine answer.

### 4.1. Organ-Aware Memory Initialization

We provide $A_\theta$ with compact organ descriptions as initial memory, offering a global overview of the CT volume for subsequent reasoning. Specifically, given a CT volume, we leverage VISTA3D to generate segmentation masks $Y_\mathcal{O}$ for major organs $\mathcal{O}$. For each organ, We compute their size, mean HU value, and its range along the z-axis using predefined functions, and use these statistics to initialize the system memory to get $\mathcal{M}_0$:

$$\mathcal{M}_0 = \{ m_o \mid o \in \mathcal{O} \}, \\ m_o = \big( S_o, \; H_o, \; [z_{\min}(o), z_{\max}(o)] \big). \quad (1)$$

Here, $S_o$ and $H_o$ denote the organ size and mean Hounsfield units (HU) value for organ $o$. With the organ information, the MLLM agent can handle basic measurement questions and derive key evidence and quantitative indicators that support broader reasoning tasks. Notably, we do not perform lesion segmentation to inject lesion-related information into $\mathcal{M}_0$. The substantial variability in lesion definitions and label taxonomies across dataset may result in noisy lesion masks, and may introduce inaccurate cues that mislead the agent. In contrast, organ segmentation is comparatively

standardized and robust, making organ-level priors a more reliable foundation for memory initialization.

### 4.2. Coarse-to-Fine Lesion Targeting

To address lesion-related queries, we introduce a Coarse-to-Fine Lesion Targeting (CFLT) strategy that progressively refines the search space, narrowing it from the entire volume to a small set of high-confidence candidate regions. The process is implemented with CT-CLIP (Hamamci et al., 2024a), a pretrained dual-encoder that aligns 3D CT volumes and clinical descriptions. Given a CT volume $V \in \mathbb{R}^{H \times W \times D}$, the 3D image encoder $f_{\text{img}}$ outputs a dense feature map $F = f_{\text{img}}(V) \in \mathbb{R}^{h \times w \times d \times n}$, where $h, w, d$ denote the spatial resolution and $n$ is the embedding dimension. A text prompt $p$ (e.g., describing a lesion concept) is encoded by $f_{\text{txt}}$ into $\mathbf{t} = f_{\text{txt}}(p) \in \mathbb{R}^n$. In the standard volume-level classification, CT-CLIP forms a global visual embedding by averaging $F$ over spatial locations and computes its cosine similarity with $\mathbf{t}$. To obtain a finer-grained lesion ROI, we use the local embedding $\mathbf{f}_{i,j,k} \in \mathbb{R}^n$ at spatial location $(i, j, k)$ in $F$. We then $\ell_2$-normalize the local features and the text embedding to obtain $\hat{\mathbf{f}}{i, j, k}$ and $\hat{\mathbf{t}}$, and compute their cosine similarity to form a dense similarity heatmap:

$$\mathcal{H}_{i,j,k} = \hat{\mathbf{f}}_{i,j,k}^\top \hat{\mathbf{t}}, \qquad \mathcal{H} \in \mathbb{R}^{h \times w \times d}. \quad (2)$$

The resulting 3D heatmap $\mathcal{H}$ highlights locations aligned with the lesion prompt. Candidate lesion ROIs and informative slices are selected then utilizing $\mathcal{H}$ and memory $\mathcal{M}_0$:

**(1) Organ-aware filtering.** $A_\theta$ first infers the target organ

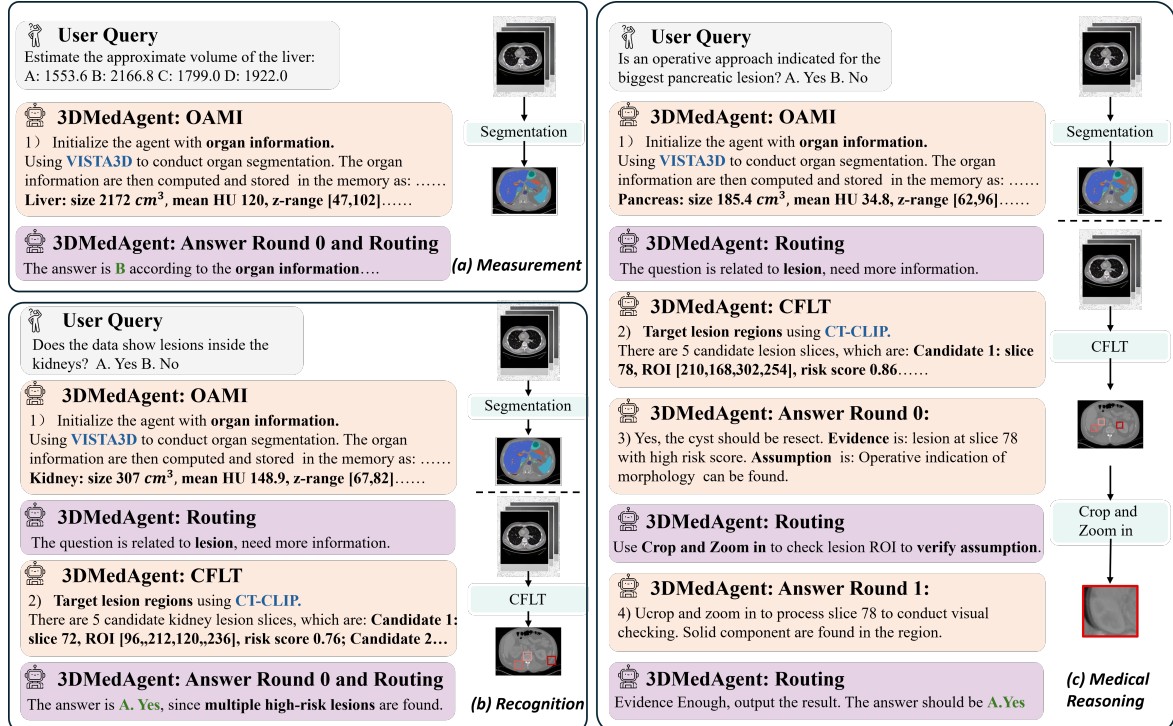

*Figure 3.* 3DMedAgent workflow. The agent answers 3D CT queries by progressively gathering and reusing evidence in shared memory, illustrated on (a) measurement, (b) recognition, and (c) medical reasoning tasks.

$o$ from the query and leverages the initialized memory $\mathcal{M}_0$ to filter out irrelevant regions. The search is constrained to $[z_{\min}(o), z_{\max}(o)]$, pruning heatmap responses outside this range to obtain an organ-masked heatmap $\mathcal{H}^{(o)}$.

**(2) ROI scoring for adaptive targets.** Depending on the query, the candidate ROIs $\mathcal{R}$ may be a set of axial slices or organ sub-regions (e.g., liver segments). We project $\mathcal{R}$ onto the heatmap grid to get $\Pi(\mathcal{R})$ and compute its score by aggregating the patch-level responses. Let $Y_o$ denote the organ mask and $\mathcal{H}(P)$ the heatmap response for patch $P$. We first discard low-response patches with $\mathcal{H}(P) < \tau$. For each remaining patch, we compute its organ overlap ratio $\rho(P)$ by projecting $Y_o$ onto the heatmap and measuring the voxels inside $P$ that fall within the organ. The lesion-related metric for each ROI is then computed as:

$$S(\mathcal{R}) = \sum_{P \subset \Pi(\mathcal{R})} \mathbb{I}[\mathcal{H}(P) \geq \tau] \, \rho(P)\mathcal{H}(P). \quad (3)$$

We rank candidate ROIs by $S(\mathcal{R})$ and add the top-scoring ones into $\mathcal{M}_0$ as lesion candidates to get $\mathcal{M}_\ell$. This yields a compact shortlist of informative ROI slices or regions that guides subsequent agentic verification and reasoning.

### 4.3. Think-with-1-Slice Loop

Some questions still require fine-grained visual inspection

---

**Algorithm 1** The iterative process of T1S-Loop.

1: **Input:** query $q$, memory $\mathcal{M}$, tool set $\mathcal{T}$, maximum iterations $T_{\max}$
2: **Output:** final answer $\hat{y}$
3: **for** $t = 0$ to $T_{\max}$ **do**
4:    $(r_t, \hat{y}_t, \mathcal{E}_t, \mathcal{A}_t) \leftarrow A_\theta(q, \mathcal{M})$
5:    $(b_t, \tau_t) \leftarrow \mathcal{R}_\theta(r_t, \hat{y}_t, \mathcal{E}_t, \mathcal{A}_t, \mathcal{T})$
6:    **if** $b_t = 0$ **then**
7:       **return** $\hat{y}_t$
8:    **end if**
9:    $s_t \leftarrow \text{SELECTSLICE}(\mathcal{M}, \mathcal{A}_t)$
10:   $v_t \leftarrow \text{APPLYTOOL}(\tau_t, s_t)$
11:   $\mathcal{M} \leftarrow \mathcal{M} \cup \{r_t, \hat{y}_t, \mathcal{E}_t, \mathcal{A}_t, s_t, v_t\}$
12: **end for**
13: **return** $\hat{y}_{T_{\max}}$

---

to resolve remaining ambiguities, discover missing visual evidence, and correct potential errors after OAMI and CFLT. To this end, we propose a Think-with-1-Slice Loop (T1S-Loop), where $A_\theta$ iteratively selects informative slices according to the current memory and conducts tool-assisted visual reasoning to progressively refine the answer. At each turn, the agent first makes a basic decision from the query and memory, including its rationale, current answer, supporting evidence, and explicit assumptions. A router then decides whether the current evidence is sufficient. If not, the

*Table 2.* The comparison results on the DeepTumorVQA benchmark. The "Rand" column indicates the random accuracy of the corresponding question subtype. Best results are **bold** and second best results are underlined.

| Type | Subtype | Rand | GPT-5 | Qwen3-VL (30B) | MedGemma (27B) | HuatuoGPT (34B) | M3D | RadFM | CT-Chat | 3DMedAgent (GPT-5) | 3DMedAgent (Qwen3-VL) |
|---|---|---|---|---|---|---|---|---|---|---|---|
| Measurement | lesion volume measurement | 0.25 | 0.25 | 0.30 | 0.24 | 0.20 | 0.18 | 0.25 | 0.17 | **0.42** | 0.35 |
| | organ HU measurement | 0.25 | 0.35 | 0.35 | 0.21 | 0.35 | 0.33 | 0.37 | 0.22 | **0.82** | 0.78 |
| | organ volume measurement | 0.25 | 0.39 | 0.41 | 0.41 | 0.24 | 0.32 | 0.31 | 0.25 | **0.63** | **0.63** |
| | Average | 0.25 | 0.33 | 0.35 | 0.29 | 0.26 | 0.28 | 0.31 | 0.21 | **0.63** | 0.59 |
| Recognition | colon lesion existence | 0.50 | 0.47 | 0.47 | 0.50 | 0.57 | 0.60 | 0.48 | 0.53 | **0.82** | 0.80 |
| | kidney cyst existence | 0.50 | 0.57 | 0.48 | 0.50 | 0.57 | 0.57 | 0.48 | 0.53 | **0.82** | 0.78 |
| | kidney lesion existence | 0.50 | 0.46 | 0.47 | 0.49 | 0.56 | 0.44 | 0.41 | 0.51 | **0.63** | 0.55 |
| | kidney tumor existence | 0.50 | 0.58 | 0.52 | 0.57 | 0.53 | 0.52 | 0.55 | 0.48 | **0.85** | 0.82 |
| | liver lesion existence | 0.50 | 0.38 | 0.47 | 0.50 | 0.43 | 0.60 | 0.43 | 0.50 | 0.73 | **0.75** |
| | pancreatic lesion existence | 0.50 | 0.53 | 0.53 | 0.49 | 0.60 | 0.49 | 0.49 | 0.51 | **0.72** | 0.68 |
| | Average | 0.50 | 0.49 | 0.51 | 0.51 | 0.54 | 0.53 | 0.47 | 0.51 | **0.76** | 0.73 |
| Visual Reasoning | adjacent organ | 0.33 | 0.43 | 0.45 | 0.60 | 0.60 | 0.53 | 0.55 | **0.62** | 0.45 | 0.42 |
| | inter-segment comparison | 0.33 | 0.38 | 0.30 | 0.40 | 0.50 | 0.23 | 0.23 | 0.52 | **0.53** | 0.45 |
| | kidney volume comparison | 0.33 | 0.30 | 0.27 | 0.37 | 0.35 | 0.32 | 0.38 | 0.32 | **0.70** | 0.62 |
| | largest lesion attenuation | 0.33 | 0.30 | 0.38 | 0.37 | 0.33 | 0.30 | 0.32 | 0.28 | 0.45 | **0.47** |
| | largest lesion diameter | 0.25 | 0.32 | 0.17 | 0.28 | 0.22 | 0.12 | 0.10 | 0.10 | **0.43** | 0.40 |
| | largest lesion location | 0.42 | 0.40 | 0.52 | 0.37 | 0.53 | 0.53 | 0.43 | 0.42 | **0.60** | 0.55 |
| | largest lesion slice | 0.25 | 0.40 | 0.20 | 0.23 | 0.27 | 0.15 | 0.15 | 0.15 | **0.52** | 0.43 |
| | lesion count by location | 0.25 | 0.42 | 0.43 | 0.32 | 0.22 | 0.20 | 0.18 | 0.18 | 0.45 | **0.48** |
| | lesion counting | 0.33 | 0.39 | 0.56 | 0.62 | 0.48 | 0.44 | 0.45 | 0.10 | **0.66** | 0.65 |
| | lesion outlier | 0.50 | 0.50 | 0.50 | 0.47 | 0.43 | 0.47 | 0.55 | 0.47 | **0.65** | 0.55 |
| | liver lesion clustering | 0.33 | 0.42 | 0.42 | 0.37 | 0.35 | 0.40 | 0.33 | 0.37 | **0.43** | 0.38 |
| | organ aggregation | 0.25 | 0.36 | 0.27 | 0.22 | 0.17 | 0.24 | 0.14 | 0.24 | 0.55 | **0.58** |
| | organ enlargement | 0.50 | 0.50 | 0.37 | 0.45 | 0.42 | 0.53 | 0.38 | 0.24 | **0.88** | 0.82 |
| | tumor organ HU difference | 0.28 | 0.40 | 0.23 | 0.30 | 0.20 | 0.23 | 0.32 | 0.17 | **0.52** | 0.47 |
| | Average | 0.34 | 0.39 | 0.35 | 0.38 | 0.36 | 0.34 | 0.34 | 0.31 | **0.56** | 0.52 |
| Medical Reasoning | fatty liver | 0.33 | 0.30 | 0.30 | 0.30 | 0.35 | 0.27 | 0.28 | 0.37 | **0.77** | 0.63 |
| | lesion type classification | 0.50 | 0.44 | 0.63 | 0.43 | 0.47 | 0.44 | 0.30 | 0.28 | **0.80** | 0.73 |
| | pancreatic cyst resectability | 0.50 | 0.50 | **0.58** | 0.53 | 0.47 | 0.52 | 0.52 | 0.50 | 0.58 | 0.52 |
| | pancreatic lesion resectability | 0.33 | 0.30 | 0.18 | 0.23 | 0.18 | 0.52 | 0.32 | 0.35 | **0.72** | 0.62 |
| | pancreatic steatosis | 0.50 | 0.65 | 0.65 | 0.47 | 0.47 | 0.52 | 0.53 | 0.48 | **0.77** | 0.60 |
| | pancreatic tumor staging | 0.25 | 0.40 | 0.37 | 0.13 | 0.20 | 0.20 | 0.25 | 0.15 | **0.53** | 0.43 |
| | Average | 0.40 | 0.43 | 0.45 | 0.35 | 0.36 | 0.42 | 0.37 | 0.36 | **0.70** | 0.59 |
| Total Average | | 0.37 | 0.41 | 0.42 | 0.38 | 0.38 | 0.39 | 0.37 | 0.35 | **0.66** | 0.61 |

agent selects one unvisited slice or ROI, applies a visual operation such as mask overlay or crop-and-zoom, and updates the memory with the newly acquired evidence. The loop terminates when the evidence is sufficient or the maximum number of iterations is reached. Formally, at iteration $t$, the agent produces

$$(r_t, \hat{y}_t, \mathcal{E}_t, \mathcal{A}_t) = A_\theta(q, \mathcal{M}), \qquad (4)$$

where $r_t$ is a brief rationale, $\hat{y}_t$ is the current answer, $\mathcal{E}_t$ lists the supporting evidence from $\mathcal{M}$, and $\mathcal{A}_t$ lists assumptions caused by missing information. The router outputs

$$(b_t, \tau_t) = \mathcal{R}_\theta(r_t, \hat{y}_t, \mathcal{E}_t, \mathcal{A}_t, \mathcal{T}), \qquad (5)$$

where $b_t \in \{0, 1\}$ indicates whether to acquire new evidence, and $\tau_t \in \mathcal{T} \cup \{\varnothing\}$ specifies the selected operation. If $b_t = 1$, the selected tool is applied to one slice or ROI $s_t$, producing visual evidence $v_t$ and updating memory:

$$\mathcal{M} \leftarrow \mathcal{M} \cup \{r_t, \hat{y}_t, \mathcal{E}_t, \mathcal{A}_t, s_t, v_t\}. \qquad (6)$$

## 5. Experiments

### 5.1. Experimental Setup

**Benchmarks.** For a comprehensive evaluation, we adopt **DeepTumorVQA** (Chen et al., 2025b) and our constructed **DeepChestVQA** benchmark. Notably, for DeepTumorVQA, we curate a refined subset by removing semantically redundant questions and balancing the distribution of answer content.(Details in Appendix B). The refined subset contains 1,740 VQA pairs drawn from 1,280 CT volumes.

**Comparative Methods.** To facilitate a comprehensive comparison, we benchmark 3DMedAgent against general MLLMs (GPT-5 (Achiam et al., 2023) and Qwen3-VL-30B (Yang et al., 2025)), 2D medical MLLMs (MedGemma-27B (Sellergren et al., 2025) and HuatuoGPT-Vision-34B (Chen et al., 2024a)), and 3D-specialized MLLMs (RadFM (Wu et al., 2025) and M3D (Bai et al., 2024)).

**Implementation Details.** MLLMs and visual models are kept zero-shot in all experiments. $T_{\max}$ are set 5 in T1S-Loop. For general and medical MLLMs, we resample each volume to a depth of 60 and uniformly sample 10 slices as

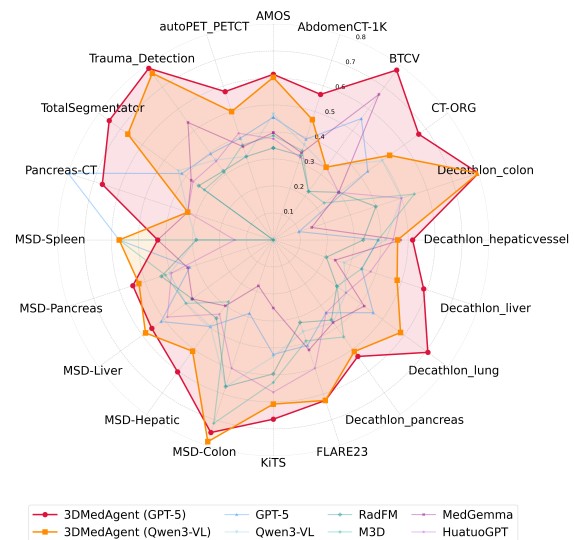

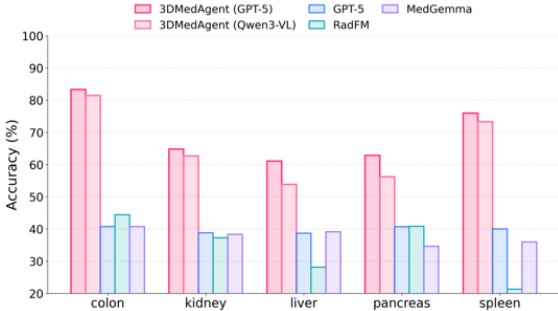

*Figure 5.* Performance comparison across different organs.

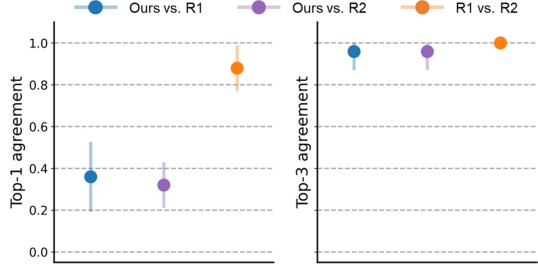

*Figure 4.* Performance across source datasets in DeepTumorVQA.

*Figure 6.* Slice-selection agreement with radiologists.

the input. We report MCQ accuracy in all experiments.

## 5.2. Comparison with MLLMs

**Comparison Results on DeepTumorVQA dataset.** As shown in Table 2, we compare 3DMedAgent across 29 question subtypes spanning four question types, where: (i) General and medical MLLMs perform poorly, showing little improvement over random guess. This is because they are primarily designed for 2D visual inputs, and thus fail to capture volumetric information. Surprisingly, even 3D-specific MLLMs underperform, suggesting that they may heavily overfit to specific datasets during fine-tuning and lack general 3D understanding. (ii) In contrast, 3DMedAgent achieves substantial gains on nearly all tasks. Using GPT-5 as the agent, 3DMedAgent delivers over 20% accuracy gains across all four question types, surpassing all baselines by substantial margins. Notably, on the most challenging medical reasoning tasks, the improvement is even larger, exceeding 27%, illustrating the superior of our design. (iii) When using Qwen3-VL as the agent, performance decreases slightly compared to GPT-5, but 3DMedAgent still outperforms other baselines. Notably, Qwen3-VL and GPT-5 yield similar results on *measurement* and *recognition*, indicating that our gains are robust and do not rely on a specific 2D MLLM choice. The larger gap arises in *medical reasoning*, likely due to Qwen3-VL's limitation in evidence integration for reaching correct conclusions.

**Comparison Results on DeepChestVQA.** Table 3 illustrates results on DeepChestVQA. Similar to the findings on DeepTumorVQA, all competing methods perform poorly, while 3DMedAgent remains strong and achieves consistent improvements, demonstrating robust generalization from abdominal to thoracic CT. Although Qwen3-VL is stronger

than GPT-5 as a medical-reasoning baseline, 3DMedAgent performs better with GPT-5, suggesting that evidence integration is the more critical for agents in our framework. Notably, improvements on *medical reasoning* are minor than on DeepTumorVQA, consistent with the diagnostic complexity and heterogeneity of thoracic CT interpretation (Mei et al., 2023), which further underscores DeepChestVQA as an important benchmark for chest CT analysis.

## 5.3. Generalization Ability Evaluation

**Cross-Dataset Generalization on DeepTumorVQA.** DeepTumorVQA is curated from various public available datasets (Details in Appendix B). Figure 4 reports dataset-wise accuracy for each method, enabling a direct generalization comparison. Fine-tuned medical or 3D-specific MLLMs exhibit unstable performance, with sharp drops on specific sources (e.g., RadFM on MSD-Spleen (Antonelli et al., 2022)). In contrast, 3DMedAgent adaptively seeks task-relevant evidence, showing great generalization across various domains.

**Cross-Organ Generalization:** Figure 8 shows the performance across organs in both abdominal and thoracic regions (Full results in Appendix C.1). 3DMedAgent consistently achieves the best results across all organs, highlighting its strong cross-organ robustness and its potential as a unified solution for general-purpose 3D medical image analysis.

## 5.4. Ablation Study on Key Components

To assess the contribution of each component, we conduct ablation studies on both datasets. As shown in Table 4, we incrementally build the full system by adding OAMI, CFLT,

*Table 3.* The comparison results on the DeepChestVQA benchmark. The "Rand" column indicates the random accuracy of the corresponding question subtype. Best results are **bold** and second best results are underlined.

| Type | Subtype | Rand | GPT-5 | Qwen3-VL (30B) | MedGemma (27B) | Huatuo-GPT (34B) | M3D | RadFM | 3DMedAgent (GPT-5) | 3DMedAgent (Qwen3-VL) |
|---|---|---|---|---|---|---|---|---|---|---|
| Recognition | bronchus lesion existence | 0.50 | 0.53 | 0.55 | 0.50 | 0.55 | 0.58 | 0.45 | 0.63 | **0.65** |
| | lung lesion existence | 0.50 | 0.52 | 0.53 | 0.50 | 0.50 | 0.52 | 0.48 | **0.68** | 0.65 |
| | pleura lesion existence | 0.50 | 0.50 | 0.50 | 0.50 | 0.55 | 0.45 | 0.48 | **0.77** | 0.73 |
| | Average | 0.50 | 0.52 | 0.53 | 0.50 | 0.53 | 0.52 | 0.47 | **0.69** | 0.68 |
| Visual Reasoning | largest lesion diameter | 0.25 | **0.38** | 0.27 | 0.10 | 0.27 | 0.22 | 0.28 | **0.38** | 0.32 |
| | largest lesion location | 0.20 | 0.18 | 0.38 | 0.23 | 0.27 | 0.17 | 0.20 | 0.38 | **0.40** |
| | largest lesion slice | 0.25 | 0.30 | 0.25 | **0.40** | 0.12 | 0.17 | 0.20 | **0.40** | 0.33 |
| | lesion count by location | 0.25 | 0.55 | 0.27 | 0.27 | 0.23 | 0.33 | 0.35 | **0.58** | 0.52 |
| | lesion counting | 0.25 | 0.38 | 0.13 | 0.20 | 0.18 | 0.30 | 0.22 | **0.45** | 0.42 |
| | organ enlargement | 0.50 | 0.38 | 0.50 | 0.50 | 0.52 | 0.58 | 0.50 | **0.63** | **0.63** |
| | organ atrophy | 0.50 | 0.55 | 0.45 | 0.52 | 0.55 | 0.50 | 0.43 | **0.60** | 0.57 |
| | lesion organ HU difference | 0.25 | 0.45 | 0.32 | 0.23 | 0.15 | 0.27 | 0.23 | **0.52** | 0.43 |
| | Average | 0.31 | 0.38 | 0.32 | 0.31 | 0.29 | 0.32 | 0.30 | **0.49** | 0.45 |
| Medical Reasoning | attenuation pattern classification | 0.50 | 0.50 | 0.57 | 0.38 | 0.48 | 0.43 | 0.50 | 0.60 | **0.62** |
| | volume-loss lesion classification | 0.50 | 0.50 | 0.48 | 0.50 | 0.32 | 0.50 | 0.42 | **0.62** | 0.58 |
| | imaging phenotype analysis | 0.25 | 0.25 | 0.32 | 0.25 | 0.30 | 0.23 | 0.23 | **0.47** | 0.43 |
| | phenotype mixing identification | 0.50 | 0.48 | 0.50 | 0.53 | 0.53 | 0.50 | 0.53 | 0.57 | **0.58** |
| | emphysema severity grading | 0.33 | 0.30 | 0.43 | 0.36 | 0.32 | 0.28 | 0.15 | 0.42 | **0.45** |
| | pleural effusion grading | 0.33 | 0.30 | 0.42 | 0.42 | 0.32 | 0.30 | 0.27 | **0.52** | 0.50 |
| | Average | 0.40 | 0.39 | 0.45 | 0.41 | 0.40 | 0.38 | 0.35 | **0.53** | 0.52 |
| Total Average | | 0.40 | 0.43 | 0.43 | 0.41 | 0.41 | 0.41 | 0.37 | **0.57** | 0.55 |

*Table 4.* Ablation on our three key components on the DeepTumorVQA and DeepChestVQA datasets. "I." and "T." are short of Image and Text, the second and the third settings only accept textual input. GPT-5 is utilized as a baseline in the ablation experiment.

| Setting | | | MLLM Input | DeepTumorVQA | | | | DeepChestVQA | | |
|---|---|---|---|---|---|---|---|---|---|---|
| OAMI | CFLT | T1S-Loop | Text / Image | Mea. | Rec. | Visual Reason. | Medical Reason. | Rec. | Visual Reason. | Medical Reason. |
| | | | I. + T. | 0.33 | 0.49 | 0.38 | 0.43 | 0.52 | 0.38 | 0.39 |
| ✓ | | | T. | 0.54 | 0.49 | 0.43 | 0.62 | 0.51 | 0.44 | 0.45 |
| ✓ | ✓ | | T. | 0.60 | 0.73 | 0.54 | 0.66 | 0.67 | 0.46 | 0.50 |
| ✓ | ✓ | ✓ | I. + T. | **0.63** | **0.76** | **0.56** | **0.70** | **0.69** | **0.49** | **0.53** |

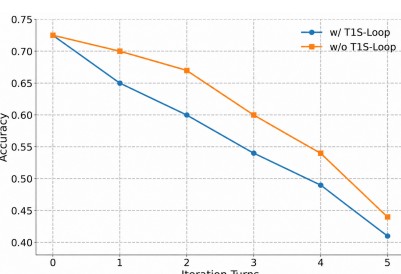

*Figure 7.* Analysis of T1S-Loop's gain across iteration turns.

and T1S-Loop in sequence, and each module yields consistent gains. OAMI provides compact organ-level priors that give the MLLM a brief overview of the 3D volume, leading to a large improvement on both perception-based *measurement* tasks and *reasoning*-related tasks. Building on OAMI, CFLT further supplies lesion information, which benefits lesion-related *recognition* and *visual reasoning* tasks significantly. Finally, T1S-Loop leverage current memory and available tools to conduct slice-level verification, refining uncertain cases and delivering the best overall performance.

## 5.5. Detailed Analysis of CFLT

We assess whether the slices selected by our agent align with expert preference. For each case, CFLT ranks all slices and outputs top-1 and top-3 candidates, while two radiologists (R1, R2) independently annotate their preference. We report agreement with each radiologist, inter-radiologist agreement, and mean±std aggregated across organs. Figure 6 shows high top-3 agreement with radiologists, approaching inter-radiologist agreement, suggesting that the agent consistently identifies clinically representative slices. Top-1 agreement is lower but remains competitive. The high inter-radiologist agreement at both top-1 and top-3 further supports the reliability of this evaluation protocol.

## 5.6. Detailed Analysis of T1S-Loop

As shown in Figure 7, performance drops markedly as the iteration turns increases, indicating that the router indeed allocates more iterations to harder cases. Enabling T1S-Loop consistently improves accuracy at every turn, with the largest gains in the first few iterations, suggesting that a

small amount of slice-level verification efficiently resolves ambiguities left by OAMI and CFLT. The gains diminish in later turns, implying that the remaining cases demand medical understanding beyond tcurrent MLLMs' capabilities.

## 6. Conclusion

This paper proposes 3DMedAgent, an 3D medical imaging analysis agent that unifies perception and understanding. 3DMedAgent provides a scalable paradigm for 3D clinical assistants: shifting from training specialized 3D models to building agents that can actively acquire and validate evidence. We hope this work will motivate future advances in stronger medical understanding, richer tool suites, and adaptive learning for reliable 3D medical decision support.

## Acknowledgements

This work was supported by Ministry of Education Tier 2 grant, Singapore (T2EP20224-0028) and Ministry of Education Tier 1 grant, Singapore (24-1250-P0001).

## Impact Statement

This paper advances AI-assisted 3D medical image analysis by enabling scalable, evidence-based reasoning over volumetric scans, which may support future clinical decision support and help reduce the burden of manual image review. While this contributes positively to clinical decision support, we acknowledge that any AI system deployed in healthcare must undergo strict clinical validation and operate under human supervision to prevent potential misuse or over-reliance on automated diagnoses.

**Conflict of Interest Disclosure.** The authors declare no financial or non-financial conflicts of interest related to this work.

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

# Appendix

**Table of content:**

## A. Details of DeepChestVQA Dataset Construction

Here we provide detailed information on how we construct the DeepChestVQA dataset detailedly, and introduce the definition and construction process of each question subtypes.

### A.1. Recognition Question Type

**Bronchus Lesion Existence:** Bronchus Lesion Existence determines whether a case contains any airway/bronchus-related abnormality by asking a binary question ("present" vs. "absent") based on lesions whose anatomical region is manually defined as bronchus/airway. This task is medically meaningful because bronchial wall thickening and bronchiectasis are key CT findings that indicate airway involvement and are routinely reported as they characterize obstructive or chronic airway disease phenotypes. We generate this subtype without loading the CT volume: for each image ID, we aggregate its disease-label set and assign the ground-truth as positive if it contains at least one label mapped to the bronchus/airway region, and negative otherwise;

**Lung Lesion Existence:** Lung Lesion Existence is a binary task that determines whether a case contains any lung-parenchymal abnormality by checking for the presence of lesions whose anatomical region is manually defined as lung in our label–region mapping. This task is medically meaningful because identifying whether the lung parenchyma is involved is a fundamental first step in thoracic imaging interpretation and provides a coarse but clinically relevant stratification of cases before more detailed pattern or severity reasoning. We generate this subtype without loading the CT volume: for each image ID, we aggregate its disease-label set and assign the ground-truth as positive if it contains at least one label mapped to the lung region and negative otherwise.

**Pleura Lesion Existence:** Pleura Lesion Existence is a binary task that determines whether a case shows any pleural abnormality by checking for the presence of lesions whose anatomical region is manually defined as pleura in our label–region mapping. This task is medically meaningful because pleural processes such as pleural effusion or pleural thickening are common and clinically important CT findings that can indicate extra-parenchymal disease and may significantly affect lung expansion. We generate this subtype without loading the CT volume: for each image ID, we aggregate its disease-label set and assign the ground-truth as positive if it contains at least one label mapped to the pleura region, and negative otherwise.

### A.2. Visual Reasoning Question Type

**Largest Lesion Diameter:** Largest Lesion Diameter measures lesion size by asking for the maximum in-plane diameter of the largest lesion present in a case, where the lesion can belong to any abnormality category provided by our annotations. This task is medically meaningful because lesion size is a fundamental quantitative descriptor in radiology that supports

standardized reporting, follow-up assessment, and longitudinal comparison across scans. We generate this subtype from the lesion masks without loading the CT volume: for each image ID, we consider all available lesion masks, identify the connected component with the largest physical area/volume as the target lesion, then compute its largest 2D diameter as the maximum Euclidean distance between any two boundary points on the slice where the component has maximal cross-sectional area; this diameter is converted to millimeters using the image spacing and used as the ground-truth value.

**Largest Lesion Location:** Largest Lesion Location determines which lung lobe contains the dominant lesion by assigning the largest lesion to one of five anatomical regions: left upper lobe, left lower lobe, right upper lobe, right middle lobe, or right lower lobe. This task is medically meaningful because lobe-level localization is a standard component of thoracic CT reporting and supports structured description, follow-up comparison, and downstream clinical decision workflows that depend on accurate anatomic attribution. We generate this subtype from masks only: for each image ID, we first identify the target lesion as the connected component with the largest physical area/volume across all lesion masks, then compute its overlap with each lobe mask and assign the ground-truth location to the lobe with the maximum overlap ratio (ties broken by absolute overlap volume/area), formatted as a 5-way multiple-choice answer.

**Largest Lesion Slice:** Largest Lesion Slice identifies where the largest lesion appears along the cranio-caudal axis by predicting the percentile position of the slice that contains the maximal lesion extent, providing a simple but clinically meaningful localization cue. This task matters because radiologists often localize the dominant abnormality by its relative position in the scan (e.g., upper vs. lower, or approximate level), which supports efficient navigation, follow-up comparison, and structured reporting. We generate this subtype using lesion masks only (without loading the CT volume): for each image ID, we scan all slices, compute the lesion area per slice (summing across all lesion masks or using the target lesion selected by maximal burden), select the slice index $z^*$ with the largest lesion area, and convert it to a percentage of the full scan depth; the ground truth is the option whose percentile value is closest to $p$ in the multiple-choice list.

**Lesion Counting:** Lesion Counting quantifies the number of discrete pulmonary nodules/masses in a case by counting how many separate lesion instances are present, focusing specifically on the nodule/mass category. This task is medically meaningful because nodule burden (how many nodules are present) is a routine and clinically relevant descriptor in chest imaging, informing disease extent characterization and follow-up assessment. We generate this subtype from the nodule/mass lesion masks: for each volume, we perform 3D connected-component analysis on the nodule/mass mask to identify distinct lesion instances, optionally filter out tiny components below a minimum physical size to reduce noise, and use the resulting component count as the ground-truth value.

**Lesion Counting by Location:** Lesion Counting by Location measures the regional burden of pulmonary nodules/masses by counting how many distinct lesion instances fall within a specified lung lobe (left upper, left lower, right upper, right middle, or right lower lobe), providing a lobe-level distribution summary rather than a global count. This is medically meaningful because radiology reports often describe not only how many nodules are present but also where they are distributed, which supports structured documentation and follow-up comparisons across time. We generate this subtype using masks only: for each case, we first extract nodule/mass instances via 3D connected-component analysis on the nodule/mass lesion mask, then assign each instance to a lobe by maximizing its overlap with the five lobe masks (ties broken by absolute overlap), and finally use the number of instances assigned to the queried lobe as the ground-truth count.

**Organ Enlargement:** Organ Enlargement assesses whether the target organ (e.g., the lungs) is abnormally enlarged by comparing its segmented volume against a cohort-derived reference, capturing a global morphologic signal rather than a focal lesion appearance. This is medically meaningful because organ size/volume is a key cue in thoracic interpretation and often differentiates major disease patterns. For example, lung enlargement (hyperinflation) is strongly associated with obstructive changes such as emphysema, while reduced lung volume (volume loss/atrophy) is commonly seen with collapse-related processes (atelectasis), fibrotic/interstitial remodeling, or compressive effects from pleural abnormalities, making "volume change" a high-level indicator that complements attenuation- and lesion-based findings. We generate this subtype using organ masks for each volume, and compare the result both across all cases in the dataset, as well as medical guidelines, and format the result as a multiple-choice options.

**Organ Atrophy:** Organ Atrophy assesses whether the target organ (e.g., the lungs) is abnormally reduced in size by comparing its segmented volume against a cohort-derived reference, capturing a global morphologic signal rather than a focal lesion appearance. This is medically meaningful because organ size/volume is a key cue in thoracic interpretation and often differentiates major disease patterns. In particular, lung volume loss (atrophy-like change) is a hallmark of collapse-related processes such as atelectasis, can reflect chronic fibrotic/interstitial remodeling with traction and architectural distortion, and may also arise from compressive effects due to pleural abnormalities (e.g., large effusions), making volume

reduction an important high-level indicator that complements attenuation- and lesion-based findings. We generate this subtype using organ masks for each volume, compute the organ volume $V_{\text{organ}}$, and determine atrophy by comparing $V_{\text{organ}}$ to both the dataset-wide distribution (e.g., lower-tail percentile or negative z-score) and guideline-informed reference ranges, then format the result as multiple-choice options.

**Lesion Organ HU Difference:** Lesion Organ HU Difference measures the attenuation contrast between a lesion and its surrounding organ parenchyma by computing the HU difference between the lesion region and a reference "normal" organ region, capturing a quantitative signal of how strongly the abnormal tissue deviates from expected background. This is medically meaningful because lesion–background attenuation contrast is a key cue in thoracic interpretation and underlies many common pattern judgments: high-attenuation lesions relative to normal lung are characteristic of alveolar opacity patterns such as consolidation and can also be seen in pleural fluid/pleural thickening and solid nodules or masses, whereas markedly low-attenuation regions relative to normal lung are typical of emphysema and related hyperinflation patterns; thus, lesion–organ HU difference provides a unified, high-level discriminator that complements morphology- and distribution-based findings. We generate this subtype from the lesion masks, organ masks, and CT intensities: for each case we compute representative HU statistics (e.g., median or trimmed mean) within the lesion mask and within a "normal" organ region defined as the organ mask excluding all lesion masks, then define the contrast as $\Delta HU = HU * \text{lesion} - HU * \text{normal}$ (optionally using $|\Delta HU|$ for contrast magnitude) and discretize it into multiple-choice bins to form the ground-truth label.

### A.3. Medical Reasoning Question Type

**Attenuation Pattern Classification:** Attenuation Pattern Classification aims to categorize the predominant CT attenuation pattern of pulmonary parenchymal abnormalities. For example, whether a case is best summarized as ground-glass opacity–dominant, consolidation–dominant, or mixed—thereby capturing a core radiologic distinction beyond simple lesion presence. This distinction is medically meaningful because ground-glass opacity and consolidation are foundational thoracic imaging descriptors that correspond to different degrees of alveolar filling and loss of aeration and are routinely used in radiology reporting to characterize the nature and severity of lung involvement (Hansell et al., 2008; Austin et al., 1996). In our dataset, this subtype is generated without reading the full volume by using the available lesion masks (e.g., GGO and consolidation/atelectasis masks) together with the lung mask to compute simple attenuation statistics within the lesion region (such as HU percentiles or occupancy in predefined HU ranges), and the ground truth label is assigned deterministically: GGO-dominant if the GGO-like attenuation component predominates, consolidation-dominant if the higher-attenuation component predominates, and mixed otherwise.

**Volume-Loss Lesion Classification:** Volume-Loss Lesion Classification evaluates whether an opacity is accompanied by significant regional lung volume reduction (collapse-leaning, e.g., atelectasis) versus occurring without clear volume loss (consolidation-leaning). This is clinically meaningful because "volume loss" is a standard thoracic imaging cue used to distinguish collapse-related opacities from other parenchymal attenuation increases that can look similar on CT (Hansell et al., 2008). We generate this subtype from lesion masks and lung/segment organ masks only: for each case, we measure opacity burden within the lesion mask and compare the corresponding lung region volume (e.g., left vs. right, or upper/middle/lower segments) to define a deterministic ground-truth label indicating the answer is volume-loss dominant or not.

**Imaging Phenotype Classification:** Imaging Phenotype Analysis summarizes each case into a single high-level radiologic phenotype by aggregating multiple lower-level findings into an "impression-like" category, rather than asking for any one lesion label. Following the schema we discussed earlier, the task maps each image to one of four options: obstructive/airway, fibrotic/interstitial, alveolar opacity, or focal/peripheral. The model must perform evidence aggregation across co-existing abnormalities (e.g., emphysema and airway findings vs. fibrotic signs vs. opacity patterns vs. focal nodules/pleural-dominant changes), which mirrors how radiologists synthesize overall patterns in routine reporting. Ground truth is generated deterministically from the per–image ID disease-label set without reading the CT volume: we assign the phenotype using your rule-based label constraints (each phenotype defined by the presence of a small set of key labels and the absence of disallowed labels, with specified "allowed extras" where applicable), discard ambiguous cases that match multiple phenotypes, and preferentially sample "more typical" cases by ranking cases with multiple key labels within the same phenotype higher during selection.

**Phenotype Mixing Identification:** Phenotype Mixing Identification determines whether a case exhibits a mixed imaging phenotype. For example, whether evidence for two or more high-level phenotypes is simultaneously present rather than being well explained by a single dominant pattern. This task directly reflects the "dominant pattern with additional features" style of radiology impressions. Consistent with the phenotype definitions we used for imaging phenotype syn. (obstructive/airway,

fibrotic/interstitial, alveolar opacity, focal/peripheral), we generate ground truth purely from the per–image ID disease-label set without reading the CT volume: for each case we compute which phenotype groups are "active" based on key-label hits, label it as mixed if at least two phenotype groups are active. We also bias sampling toward clearer examples by preferring cases with more active groups and higher within-group key-label counts. This task is medically meaningful because mixed patterns are common in real-world chest CT (e.g., obstructive changes coexisting with superimposed opacities), and explicitly identifying such co-occurrence supports more faithful, clinically interpretable summary reasoning.

**Emphysema Severity Grading:** Emphysema Severity Grading measures the extent of emphysema by assigning each case to an ordinal severity level (e.g., mild/moderate/severe), thereby capturing disease burden rather than merely presence or absence. This is clinically meaningful because emphysema extent is a standard quantitative descriptor in chest CT reporting and supports severity stratification in obstructive lung disease. We generate this subtype solely from the emphysema lesion mask and the lung mask, without loading the CT volume: for each case we compute an emphysema index

$$EI = \frac{V(M_{\text{emph}} \cap M_{\text{lung}})}{V(M_{\text{lung}})}$$

and deterministically map $EI$ to discrete grades using either fixed cutoffs or dataset-calibrated quantile bins, yielding the multiple-choice ground-truth label.

**Pleural Effusion Grading:** Pleural Effusion Grading estimates effusion severity by categorizing each case into an ordinal volume-based level (e.g., small/moderate/large), reflecting the overall burden of pleural fluid rather than simple presence. This is medically meaningful because effusion extent is routinely described in thoracic imaging and directly relates to compressive effects on lung expansion and downstream clinical management. We generate this subtype using only the pleural effusion lesion mask together with the relevant anatomical mask (e.g., hemithorax or lung), without loading the CT volume: for each case we compute an effusion ratio

$$R_{\text{eff}} = \frac{V(M_{\text{eff}} \cap M_{\text{region}})}{V(M_{\text{region}})}$$

and deterministically assign the ground-truth grade by mapping $R_{\text{eff}}$ to small/moderate/large using fixed thresholds or dataset-calibrated quantile bins, producing the multiple-choice label.

## B. Details of DeepTumorVQA Dataset Usage

Since the publicly released version of the DeepTumorVQA dataset differs slightly from the descriptions in its official paper, and some of its constituent data sources (e.g., NIH collections) are not publicly accessible, we here describe the actual source datasets that are included in the DeepTumorVQA data used in our paper.

### B.1. Source Datasets

**AMOS:** AMOS (Abdominal Multi-Organ Segmentation) is a large-scale, clinical benchmark dataset designed to support the development and evaluation of automatic abdominal organ segmentation algorithms on CT and MRI scans (Ji et al., 2022). It comprises 500 CT and 100 MRI volumes collected from multi-center, multi-vendor, multi-modality, multi-phase, and multi-disease patients, offering diverse imaging conditions that reflect real clinical scenarios. Each case includes voxel-level annotations for 15 abdominal organs, enabling comprehensive training and fair comparison of segmentation methods across a wide range of targets and clinical variations.

**AbdomenCT-1K:** AbdomenCT-1K is a large and diverse abdominal CT organ segmentation dataset assembled to evaluate and improve the generalizability of deep learning models across heterogeneous clinical imaging data (Ma et al., 2021). It contains over 1,000 contrast-enhanced CT scans collected from 12 different medical centers, encompassing multiple imaging phases, scanner vendors, and disease conditions, which better reflects real-world variability compared to many earlier single-center datasets. The dataset is annotated for four major abdominal organs — liver, kidneys, spleen, and pancreas — with voxel-level ground truth labels, and it has been organized into benchmarks for fully supervised, semi-supervised, weakly supervised, and continual learning research.

**BTCV:** BTCV (Beyond the Cranial Vault) Multi-Organ Segmentation Dataset is a public abdominal CT dataset originally curated for the MICCAI 2015 segmentation challenge and widely used as a benchmark for 3D multi-organ segmentation research. It consists of volumetric CT scans of patients acquired at the Vanderbilt University Medical Center, with manual

voxel-level annotations for multiple abdominal organs including the liver, spleen, kidneys, gallbladder, stomach, pancreas, aorta, inferior vena cava, portal and splenic veins, adrenal glands, esophagus, and duodenum (some labels vary by release). The dataset typically provides around 90 CT cases with reference segmentations or a common subset of 30–50 annotated volumes for training and evaluation in segmentation studies.

**CT-ORG:** CT-ORG is a public CT imaging dataset curated for multi-organ segmentation (Rister et al., 2020), hosted on The Cancer Imaging Archive (TCIA). It contains 140 volumetric CT scans with 3D annotations for multiple organs including the lungs, bones, liver, kidneys, and urinary bladder, and in a minority of cases also the brain. The scans represent a mix of imaging conditions (contrast-enhanced and non-contrast CTs, standalone and PET-CT derived volumes) and include both benign and malignant disease presentations, making it a challenging benchmark for developing and evaluating multi-class segmentation algorithms.

**Decathlon and MSD Series:** In the DeepTumorVQA dataset, we observed that some CT volumes are labeled with an "MSD_*" prefix while others use a "Decathlon_*" prefix in their source identifiers. This distinction can be confusing at first glance, but inspection of the actual image source mapping shows that both naming conventions refer to subsets of the Medical Segmentation Decathlon (MSD) benchmark (Antonelli et al., 2022), which comprises multiple organ- and task-specific CT segmentation tasks. DeepTumorVQA integrates 9,262 abdominal CT volumes from 17 public datasets, including the ten heterogeneous tasks defined under the MSD challenge, and maps each image to a unified AbdomenAtlas identifier for traceability.

The "MSD-" prefix typically reflects the original task names used in the Decathlon benchmark (e.g., MSD-Colon, MSD-Pancreas), whereas the "Decathlon_" prefix appears to be an alternative naming adopted in DeepTumorVQA's internal mapping, possibly reflecting a different interpretation or processing step when harmonizing the source labels. Importantly, both prefixes point to the same underlying MSD tasks, rather than to two distinct datasets. This naming variation stems from DeepTumorVQA's dataset assembly and naming normalization process (AbdomenAtlas), rather than from fundamentally different data sources.

We retain these differences in the dataset descriptions here to respect the original paper's organization and any deeper conceptual distinctions the authors may have intended by using both naming conventions, and to facilitate precise reproducibility and cross-reference with the released dataset. Notably, these datasets includes to Decathlon_colon, Decathlon_hepaticvessel, Decathlon_liver, Decathlon_lung, Decathlon_pancreas and MSD-Colon, MSD-Hepatic, MSD-Liver, MSD-Pancreas, MSD-Spleenin the DeepTumorVQA datasets.

**FLARE23:** FLARE23 (Fast, Low-resource, and Accurate oRgan and Pan-cancer sEgmentation in Abdomen CT) is a large-scale benchmark dataset introduced as part of the MICCAI 2023 FLARE Challenge (AN et al., 2023), designed to advance robust and efficient abdominal organ and pan-cancer segmentation on 3D computed tomography (CT) scans. The dataset includes thousands of abdominal CT volumes sourced from over 30 medical centers, encompassing a wide variety of imaging protocols, scanner vendors, and clinical conditions. It is one of the largest publicly-available abdominal CT segmentation resources to date, with approximately 4,000 training cases, 100 validation cases, and 400 test cases. Among the training set, around 2,200 CT volumes come with partial organ and tumor annotations, while the remaining 1,800 volumes are unlabeled, enabling semi-supervised learning scenarios.

**KITS:** KiTS (Kidney Tumor Segmentation) refers to a series of public abdominal CT segmentation datasets and challenges focused on the automatic semantic segmentation of kidneys and kidney tumors (Heller et al., 2019). The original KiTS19 dataset, released in conjunction with the 2019 MICCAI Kidney Tumor Segmentation Challenge, contains contrast-enhanced CT scans from patients who underwent partial or radical nephrectomy for renal tumors, with manual voxel-level segmentations of kidneys and tumor regions provided for training and evaluation. A subset of 210 CT volumes was publicly released with ground truth labels to support algorithm development, while additional cases were held out for challenge evaluation.

**Pancreas-CT:** Pancreas-CT is a classical publicly available abdominal CT imaging dataset established for pancreas segmentation research (Roth et al., 2016). It originates from contrast-enhanced 3D CT scans acquired at the National Institutes of Health (NIH) Clinical Center, and is hosted on The Cancer Imaging Archive (TCIA). The dataset includes 80–82 3D CT volumes of adult subjects (typically 53 males and 27 females) scanned in the portal-venous phase approximately 70 seconds after intravenous contrast injection, with voxel-level manual pancreas annotations (segmentation masks) generated slice-by-slice and verified by radiologists. The scans have a matrix size of 512×512 pixels and slice thicknesses generally in the 1.5–2.5 mm range, acquired on Philips and Siemens MDCT scanners.

**TotalSegmentator:** TotalSegmentator is a publicly available, large-scale 3D CT anatomical segmentation dataset and toolkit

developed by researchers from the University Hospital Basel (Wasserthal et al., 2023). The original dataset (v1) consists of 1204 clinical CT examinations that were manually segmented for 104 anatomically relevant structures, covering a broad range of 27 organs, 59 bones, 10 muscles, and 8 vessels sampled from routine clinical imaging settings with diverse scanners, protocols, and pathologies. This diversity helps the dataset generalize across real-world variability in CT imaging. The dataset has since been extended (v2) to include 117 anatomical structures across 1228 CT volumes, representing one of the most comprehensive publicly released CT segmentation annotation collections to date.

**Trauma_Detection**: Trauma_Detection in DeepTumorVQA refers to CT data sourced from the RSNA Abdominal Traumatic Injury CT (RATIC) dataset (Rudie et al., 2024), which was created for the RSNA 2023 Abdominal Trauma Detection AI Challenge. The RATIC dataset is one of the largest publicly available collections of abdominal CT studies annotated for traumatic injuries, comprising thousands of scans from dozens of institutions around the world. It includes image-level and study-level annotations for traumatic injury findings, such as injuries to solid organs (e.g., liver, spleen, kidneys), bowel/mesenteric injuries, and active extravasation, along with injury grading provided by expert radiologists.

**autoPET_PETCT:** autoPET_PETCT refers to a publicly available collection of whole-body PET/CT imaging studies established in the context of the autoPET Grand Challenge on automated lesion segmentation in oncologic PET/CT data. This dataset comprises co-registered 3D positron emission tomography (PET) and computed tomography (CT) volumes acquired on state-of-the-art PET/CT scanners following standardized clinical protocols. The core cohort includes patients with histologically confirmed malignant melanoma, lymphoma, or non-small cell lung cancer (NSCLC) as well as negative control subjects without PET-avid malignant lesions, with most exams extending from the skull base to mid-thigh level as whole-body acquisitions. Each case typically consists of a 3D FDG-PET volume, a corresponding 3D CT volume, and manually segmented tumor lesion masks on the PET images aligned with the CT anatomy.

### B.2. The preprocessing details of DeepTumorVQA

Notably, the original dataset scales up by introducing multiple linguistic paraphrases for semantically identical VQA pairs, and some MCQ subtypes exhibit substantial label imbalance (e.g., in lesion type classification, *cyst* accounts for 97% of the answers). To construct a cleaner and more informative evaluation set, we adopt the following sampling and balancing strategy.

First, we aim to preserve broad coverage over data sources. Since DeepTumorVQA is curated from more than 20 public CT datasets, we sample VQA pairs in an approximately source-balanced manner, so that each source dataset contributes a comparable number of test examples. This reduces evaluation bias toward a few dominant sources and better reflects generalization across heterogeneous acquisition protocols and patient cohorts.

Second, we explicitly remove redundancy introduced by paraphrasing. For VQA pairs with the same semantic meaning, we do not repeatedly sample multiple paraphrases. Concretely, for each CT case, we keep at most one question per question subtype (i.e., for a fixed `case` and `subtype`, we sample only a single VQA instance), ensuring that the test set emphasizes question diversity rather than linguistic rewordings.

Third, we encourage case diversity by spreading questions across as many distinct CT volumes as possible, instead of allowing a small number of volumes to contribute a disproportionate number of QAs. In practice, we cap the number of sampled VQA pairs per CT volume and prioritize under-represented cases during sampling. This mitigates overly optimistic results caused by repeated querying of the same volume (which can implicitly leak case-specific cues across questions) and yields a more faithful estimate of robustness at the case level.

Finally, we address answer-content imbalance for closed-option MCQs. For open-valued or effectively open-option questions (e.g., many measurement questions whose answer options vary continuously across cases), we do not enforce balancing constraints. In contrast, for subtypes with fixed semantic option contents (e.g., tumor staging where options correspond to canonical categories such as T1–T4, even if the option letters are shuffled), we balance according to the *option content* rather than the option letter. Specifically, we sample examples so that each semantic option (e.g., T1, T2, T3, T4) appears as evenly as possible in the final subset, reducing spurious performance inflation on majority answers and making accuracy a more meaningful indicator of true capability.

Together, these procedures produce a test subset with (i) broader source coverage, (ii) reduced redundancy, (iii) higher case diversity, and (iv) improved balance for closed-option subtypes, leading to a more reliable and informative benchmark for evaluating 3D medical VQA systems.

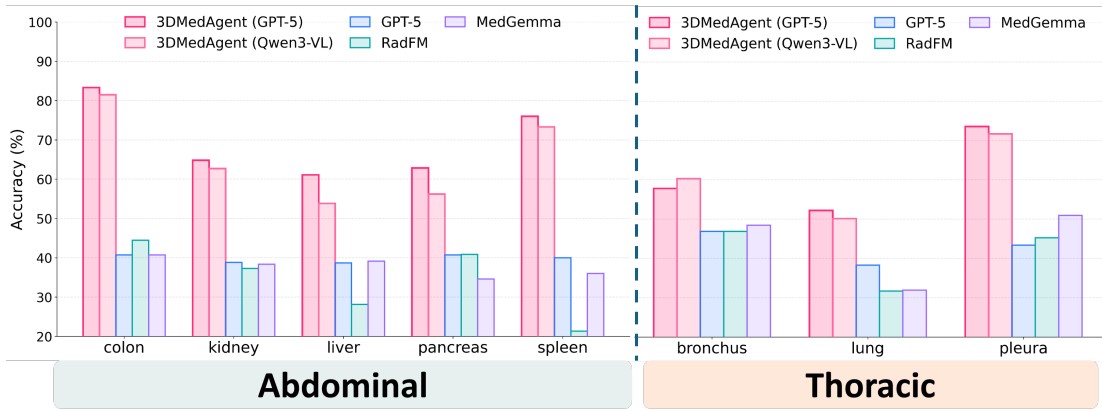

*Figure 8.* Performance comparison across different abdominal and thoracic organs.

# C. Additional Experimental Results

Here we provide additional results that cannot be present in main text dual to space limiation.

## C.1. Full results of generation across organs

*Table 5.* Organ/region distribution in DeepTumorVQA and DeepChestVQA test subsets.

| DeepTumorVQA | | DeepChestVQA | |
|---|---|---|---|
| Organ | #Samples | Region | #Samples |
| Colon | 54 | Bronchus/Airway | 114 |
| Kidney | 657 | Lung | 803 |
| Liver | 455 | Pleura | 103 |
| Pancreas | 555 | – | – |
| Spleen | 75 | – | – |

Here we present the full comparison on both abdominal and thoracic organs. In abdominal region, the organs mainly span across colon, kidney, liver, pancreas. In the thoracic region, the organs mainly span across bronchus, lung and pleura. 3DMedAgent performas well across all organs, illustrating the strong generalization ability of our method. We also present the static of organ distribution across both datasets:

## C.2. Ablation on Different Segmentation Backbone

To study the robustness of our framework to the choice of segmentation model, we replace the segmentation model in OAMI from VISTA3D with TotalSegmentator and report the results below:

*Table 6.* Ablation study on segmentation tools. We compare using VISTA3D and TotalSegmentator against the GPT-5 baseline.

| Setting | Mea. | Rec. | Visual Rea. | Medical Rea. | Overall |
|---|---|---|---|---|---|
| GPT-5 (baseline) | 0.33 | 0.49 | 0.39 | 0.43 | 0.41 |
| 3DMedAgent (VISTA3D) | 0.63 | 0.76 | 0.56 | 0.70 | 0.66 |
| 3DMedAgent (TotalSeg) | 0.58 | 0.76 | 0.55 | 0.67 | 0.64 |

Measurement performance drops when replacing VISTA3D with TotalSegmentator, likely due to differences in segmentation quality. However, recognition and reasoning tasks remain stable, suggesting that the gain mainly comes from the exploration design rather than a specific segmentation tool.

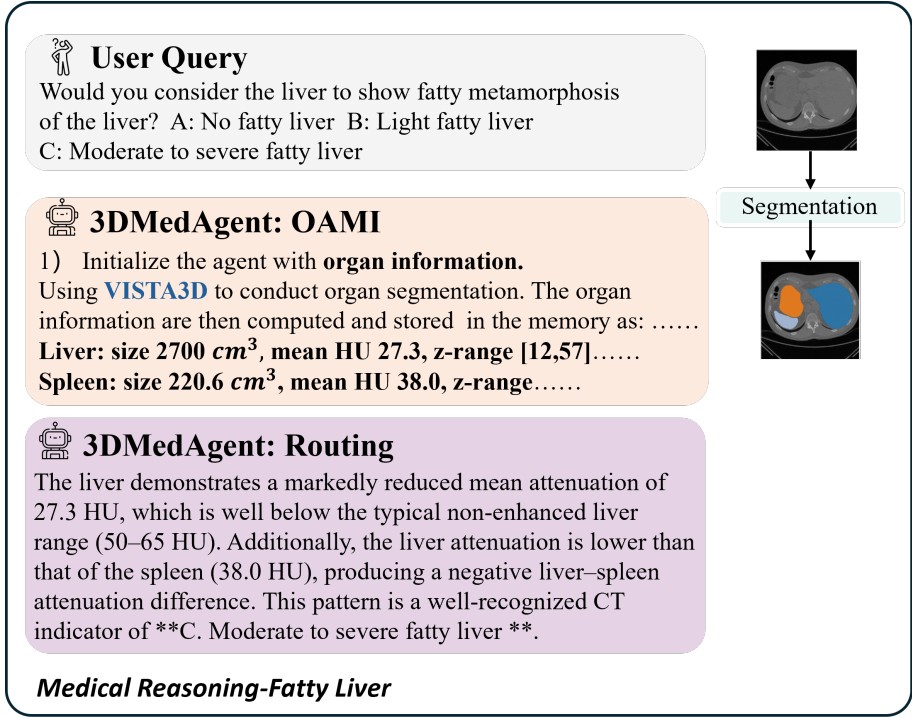

*Figure 9.* Case study on fatty liver assessment. 3DMedAgent initializes organ cues via OAMI (VISTA3D-based segmentation) and then reasons over liver–spleen attenuation evidence from shared memory to select the final diagnosis.

### C.3. Detailed Case Study

In this section, we provide additional detailed case studies to better illustrate how 3DMedAgent operates in practice. Beyond aggregate metrics, these examples visualize the agent's step-by-step evidence-seeking process illustrating how it plans tool use, accumulates and updates shared memory, and invokes slice-level verification when needed—thereby offering a more interpretable view of its decision-making and failure modes across representative 3D tasks.

As shown in Figure 9, this case study highlights that many clinical conclusions are difficult to obtain from visually inspecting a few slices alone, but can benefit substantially from low-level perception and quantitative measurement. For fatty liver assessment, reliable cues often come from organ-level measurements—such as liver volume and attenuation (HU)—rather than salient focal lesions. As shown in Fig. 9, 3DMedAgent first extracts quantitative organ evidence, including the liver HU and a reference HU from the adjacent spleen. The markedly reduced liver attenuation (and the liver–spleen contrast) provides a direct, guideline-consistent signal of hepatic steatosis, enabling the agent to identify the low-density pattern and make a confident diagnosis.

As shown in Figure 10, this case shows that measurement questions can be sensitive to upstream segmentation quality. OAMI computes the liver volume from the segmentation mask and records it in memory; however, segmentation inaccuracies bias the estimated volume. The agent then matches this erroneous measurement to the closest multiple-choice option, resulting in an incorrect selection. This failure mode suggests that reliable quantification requires either more robust segmentation, uncertainty-aware checks (e.g., plausibility ranges or mask quality scoring), or complementary measurement cues to prevent error propagation from perception to decision.

As shown in Figure 11, this case illustrates how 3DMedAgent handles measurement ambiguity through targeted verification. OAMI first computes the total lung volume from the segmentation mask and stores the estimate in memory (6035 cm³), but the value does not align with any multiple-choice option, triggering the router to invoke T1S-Loop. The agent then requests a single slice with mask overlay to inspect segmentation quality and identifies under-segmentation near the posterior costophrenic recess, which would systematically bias the volume downward. With this evidence, the agent updates its confidence and resolves the question by selecting the option consistent with the corrected volume estimate.

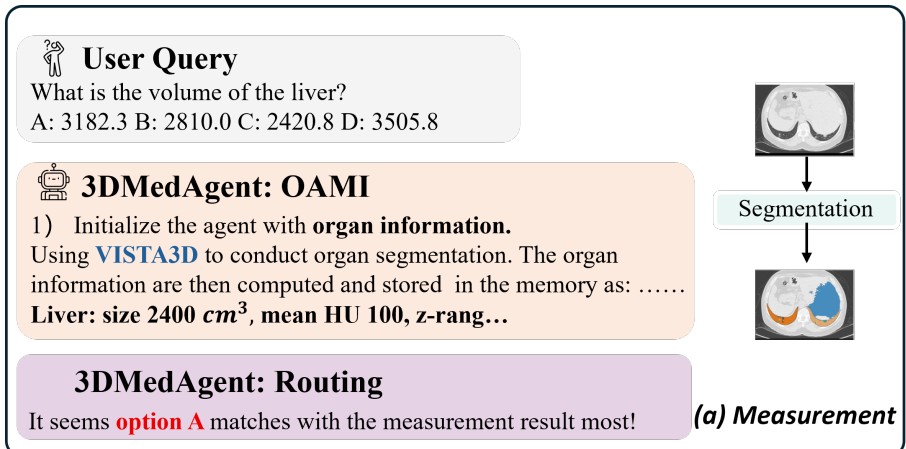

Figure 10. Failure case on organ volume measurement. A biased liver volume estimate from imperfect segmentation leads the agent to select an incorrect option.

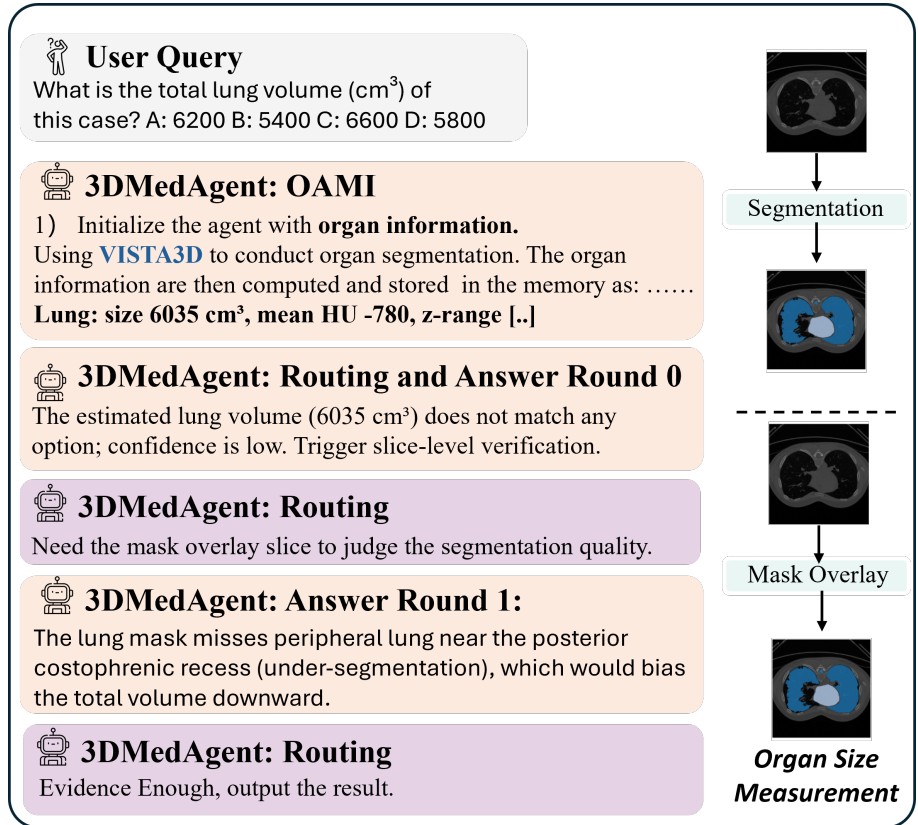

Figure 11. T1S-Loop resolves an ambiguous lung volume query via slice-level mask verification. OAMI provides an initial lung volume estimate that does not match any option, and the agent invokes one-step visual checking to assess segmentation quality before finalizing the answer.

## D. Limitations and Discussions

Despite its strong perormance, 3DMedAgent also has several limitations. First, the effectiveness of the early stages, OAMI and CFLT, is inherently bounded by the reliability and generalization of the underlying visual tools. OAMI relies on organ segmentation quality from VISTA3D, while CFLT depends on CT-CLIP for lesion localization. Errors or domain shifts in these pretrained models may propagate to downstream reasoning, leading to suboptimal evidence initialization or incomplete

lesion candidate discovery.

Second, although the T1S-Loop enables targeted visual verification, its reasoning capability is still constrained by the medical understanding of the underlying 2D MLLM. In particular, while T1S-Loop can effectively validate perceptual cues (e.g., lesion presence, relative size, or location), it remains limited in complex anatomical or relational reasoning that requires fine-grained spatial comprehension across multiple medical structures.

This limitation is reflected in certain sub-tasks where 3DMedAgent underperforms compared to strong MLLM baselines. For example, on *adjacent-organ* in the visual reasoning tasks in DeepTumorVQA (e.g., "Which organ lies adjacent to the largest pancreatic lesion?"), 3DMedAgent occasionally performs worse than general MLLMs. Such questions require precise interpretation of spatial relationships between organs, which can be challenging when reasoning from a small number of 2D slices. In contrast, baseline MLLMs may answer correctly by leveraging prior medical knowledge (e.g., typical anatomical adjacency patterns) rather than explicit visual evidence, leading to an apparent advantage in these cases. This observation highlights a fundamental trade-off in our design: 3DMedAgent emphasizes evidence-grounded reasoning over prior knowledge heuristics. While this improves robustness and interpretability for many tasks, it may be less effective when the required information is weakly observable from limited visual cues or when the task is better solved by strong anatomical priors.

Finally, our current framework adopts a zero-shot setting and does not adapt the agent policy through learning. The routing strategy, evidence aggregation, and tool-selection behaviors are fixed by prompting rather than optimized. Future work could incorporate learning-based approaches, such as supervised fine-tuning or reinforcement learning, to improve tool-use proficiency, evidence integration across iterations, and the handling of relational anatomical reasoning. We believe these directions will further strengthen the agent's ability to balance perceptual evidence with medical knowledge in complex 3D clinical scenarios.

