# OpenReview forum: "3DMedAgent: Unified Perception-to-Understanding for 3D Medical Analysis"
_ICML.cc/2026/Conference — ICML 2026 regular_

### Official Review · Reviewer_vNBv · 2026-03-08

**Soundness:** 3
**Presentation:** 2
**Significance:** 3
**Originality:** 3
**Overall Recommendation:** 5
**Confidence:** 3

**Summary:**

This paper proposes 3DMedAgent, an agentic framework that enables a 2D multimodal large language model to perform 3D medical image analysis by leveraging pretrained 3D medical models as external tools. Instead of training a dedicated 3D medical MLLM, the method decomposes 3D CT analysis into perception and reasoning steps, uses tool outputs as structured evidence, and performs multi-step reasoning with memory. The paper also introduces DeepChestVQA, a new benchmark for chest CT question answering. Overall, the paper presents an interesting and practical direction for extending strong 2D MLLMs to 3D medical imaging without requiring end-to-end 3D multimodal training.

**Compliance With Llm Reviewing Policy:**

Affirmed.

**Final Justification:**

The paper presents an interesting and practical agentic framework for 3D CT analysis with a well-designed benchmark (DeepChestVQA). All of my concerns have been adequately addressed in the rebuttal. I maintain my score of accept.

**Key Questions For Authors:**

- The paper mentions predefined functions for computing quantities such as size and mean HU value. Where can the details of these predefined functions be found? If they are not included in the paper, I encourage the authors to provide them for reproducibility.
- For the general and medical MLLM baselines, were 10 slices randomly sampled from the 60-slice volume and then provided as image inputs to the model? Please clarify the exact input construction for these baselines.
- In the T1S-Loop, why was $T_{\max}=5$ chosen? Was looking at up to five slices empirically sufficient, and was any sensitivity analysis performed for this hyperparameter?
- Figure 7 is difficult to interpret. Does it indicate that the performance without T1S-Loop is sometimes higher than with T1S-Loop? Please clarify how this figure should be read.
- In the ablation results of Table 4, the second and third rows indicate that only text is given as input to the MLLM. This requires further explanation. Does this mean that the outputs from OAMI and CFLT are converted into textual evidence and then provided to the agent without any image input, as suggested by Figure 3?

**Limitations:**

The paper does not explicitly discuss its limitations in the conclusion. I encourage the authors to include a brief discussion there, especially regarding the dependence on external pretrained 3D tools and the possible propagation of upstream tool errors into downstream reasoning.

**Strengths And Weaknesses:**

### Strengths

- The paper proposes 3DMedAgent, an agentic framework that uses a 2D MLLM together with pretrained 3D medical models as tools, which is an interesting way to address the difficulty of directly applying 2D MLLMs to 3D medical images.
- Rather than training a dedicated 3D medical MLLM, the paper presents a method built on top of existing 2D MLLMs, making the approach relatively efficient and potentially more scalable.
- The paper introduces DeepChestVQA, a new benchmark for chest CT analysis, which is a meaningful contribution because thoracic 3D medical VQA benchmarks are still limited.

### Weaknesses

- The proposed framework relies heavily on pretrained 3D medical models such as VISTA3D and CT-CLIP. As a result, the effectiveness of 3DMedAgent is closely tied to the availability and quality of such external tools, which limits the generality of the approach.
- There are several issues in the mathematical notation and presentation that make the method difficult to follow.
  - In line 213, it is unclear whether $M_0$ refers to the same object as the memory introduced earlier.
  - In Section 4.3, the variable $t$ is used without being clearly defined beforehand.
  - In the right column around line 257, $A_t$ appears without definition. It is unclear whether this denotes the agent or is a typo of $a_t$.

---

> ### Author Rebuttal · Authors · 2026-03-30
>
> Thanks for your thoughtful review. We appreciate your recognition of the paper's writing quality, clinical importance, and promising performance. We address each concern below.
>
> **W1:**
> We agree that tool dependence is an important consideration and have discussed in *Appendix D*. A key design principle of 3DMedAgent is **modularity**: perception tools are plug-and-play and can be replaced without modifying the reasoning pipeline. To validate this, we replaced VISTA3D with TotalSegmentator (another segmentation model) in OAMI and evaluated on DeepTumorVQA:
>
> | Setting               | Mea. | Rec. | Visual Rea. | Medical Rea. | Overall |
> | --------------------- | ---- | ---- | ----------- | ------------ | ------- |
> | GPT-5 (baseline)      | 0.33 | 0.49 | 0.39        | 0.43         | 0.41    |
> | 3DMedAgent (VISTA3D)  | 0.63 | 0.76 | 0.56        | 0.70         | 0.66    |
> | 3DMedAgent (TotalSeg) | 0.58 | 0.76 | 0.55        | 0.67         | 0.64    |
>
> 3DMedAgent maintains strong performance with a different tool, still substantially outperforming the baseline. The drop in *Measurement* is expected as it directly relies on segmentation quality. Reasoning tasks remain stable across the two tools, suggesting that 3DMedAgent ensures robust performance **resilient to tool variance**. We believe evaluating more diverse tool combinations is a valuable future direction and **will add to discussion**.
>
> **W2:**
> Thanks for the careful review. (1) $M_{0}$ at line 213 refers to the same organ memory in *Eq. (1)* , we will unify the notation; (2) $t$ is the iteration turn in T1S-Loop and will be defined at first use; (3) $A_t$ at line 257 is a typo for $a_t$ and will be corrected. We have reviewed the full manuscript for additional inconsistencies and will address all in the revision.
>
> **Q1:**
> These are standard statistics computed from segmentation masks: organ size is foreground voxel count × physical spacing, mean HU is the average intensity within the mask, and z-range is the axial extent. Full implementations are in our *anonymized code*. We will add explicit definitions alongside *Eq. (1)* in the revision.
>
> **Q2:**
> As stated in *Section 5.1*, we resample each volume to depth 60 and **uniformly** sample 10 slices (not randomly). This is because: (1) it eliminates randomness and ensures reproducible results; and (2) it provides uniform anatomical coverage along the entire z-axis rather than being biased to particular region. The choice of 10 slices balances coverage against context-length constraints of current MLLMs; Med-2E3 (BIBM 2025) similarly uses 8 uniformly sampled slices. 3D-specific baselines (M3D, RadFM) receive the **full 3D volume** per their original design, ensuring a fair comparison.
>
>
> **Q3:**
> T_max = 5 was validated by sensitivity analysis on DeepTumorVQA:
>
> | T_max   | 1    | 3    | 5    | 7    | 10   |
> | ------- | ---- | ---- | ---- | ---- | ---- |
> | Overall | 0.63 | 0.65 | 0.66 | 0.66 | 0.65 |
>
> Performance improves up to T_max = 5 and saturates, with slight decline at 10. This is because: (1) **Over 80% of cases are resolved within few iterations**, where the agent finds sufficient evidence and finalizes the answer; (2) the remaining cases involve ambiguity, and the MLLM lacks the medical knowledge to reach a confident decision. Therefore, these cases tend to iterate until the upper limit, and may cause second-guess on previously correct judgments. This is consistent with *Figure 7*, where later-turn cases are inherently harder. T_max = 5 balances sufficiently against over-iteration risk. **We will include this analysis in the revision.**
>
> **Q4:**
> Sincerely thanks for helping us identify a labeling error in *Figure 7*: the legends for "With T1S-Loop" and "Without T1S-Loop" are **swapped**. With corrected labels, T1S-Loop **consistently improves accuracy at every iteration turn**. The x-axis groups questions by the number of iterations the router assigns; turn 0 is easiest (answered directly), turn 5 is hardest. Both curves decline because later turns contain harder qu
> estions, not because iteration degrades performance. The consistent gap confirms T1S-Loop benefits all difficulty levels. We will correct and clarify *Figure 7* in the revision.
>
> **Q5:**
> The understanding is correct.  In *Table 4*: row 2 (OAMI only) provides organ statistics as text; row 3 (OAMI+CFLT) adds lesion candidate information in text form. Neither receives image input. Note that 3DMedAgent still performs progressive reasoning by querying memory, selecting task-relevant evidence, and integrating cues, rather than simply relying on raw outputs. We will add a clarification to the *Table 4* caption.
>
> **L1:**
> Kindly noted limitations are discussed in *Appendix D*, covering tool dependence and error propagation. *Appendix C.2 (Figure 10)* provides a concrete failure case where imperfect segmentation leads to a wrong answer. We will add a summary of key limitations to the main-text conclusion for visibility.

---

> > ### Author Rebuttal · Reviewer_vNBv · 2026-04-04
> >
> > Thank you for your detailed response. All of my questions have been addressed. I will maintain my score of accept.

---

> > > ### Author Response · Authors · 2026-04-04
> > >
> > > Thanks for your positive assessment and for your helpful questions. We are grateful that the rebuttal has further strengthened the positive assessment of the paper and reinforced your support for its acceptance. We will incorporate your suggestions further to improve the presentation and discussion in the final version.

---

### Official Review · Reviewer_B7wS · 2026-03-11

**Soundness:** 2
**Presentation:** 4
**Significance:** 2
**Originality:** 3
**Overall Recommendation:** 3
**Confidence:** 4

**Summary:**

This paper proposes 3DMedAgent, a tool-augmented framework that enables a 2D MLLM to answer 3D CT questions by orchestrating pretrained 3D visual tools and maintaining structured memory. The system consists of Organ-Aware Memory Initialization, Coarse-to-Fine Lesion Targeting, and a Think-with-1-Slice Loop that iteratively selects informative slices and updates evidence for reasoning. The paper also introduces DeepChestVQA, a chest CT benchmark with recognition, visual reasoning, and medical reasoning questions, and reports improvements over general, medical, and 3D-specific MLLMs on both DeepTumorVQA and DeepChestVQA.

**Compliance With Llm Reviewing Policy:**

Affirmed.

**Final Justification:**

The rebuttal to the weakness still does not convince me too much. I thus keep my rating

**Key Questions For Authors:**

KeyQ for authors

1：The method explicitly uses VISTA3D for organ segmentation and CT-CLIP for lesion targeting, see Sections 4.1 and 4.2, Pages 4 to 5. By contrast, the general and medical MLLM baselines are given a very weak input protocol, “resample each volume to a depth of 60 and uniformly sample 10 slices” (Section 5.1, Page 6). So Table 2 and Table 3 are effectively comparing “2D MLLMs with 10 slices” against “2D MLLM plus powerful 3D tools over the full volume.” That setup may show the usefulness of tool augmentation, but it does not justify the stronger narrative that the agent itself is the main driver of the improvement. A more convincing evaluation would include tool-aware baselines, for example a direct pipeline that feeds the same VISTA3D and CT-CLIP outputs into a simple prompt, or even a non-agent rule-based solver where possible.

2：In the main paper, DeepChestVQA is built from organ and lesion masks, with “relevant indicators computed from the masks to derive answers” for medical reasoning questions, see Section 3, Page 3. Meanwhile, OAMI computes organ size, mean HU, and z-range directly from segmentation masks, and CFLT localizes lesion ROIs using CT-CLIP heatmaps constrained by organ memory, see Sections 4.1 to 4.2, Pages 4 to 5. Table 4 on Page 8 makes this especially visible: adding OAMI alone, with text-only MLLM input, boosts DeepTumorVQA medical reasoning from 0.43 to 0.62 and DeepChestVQA visual reasoning from 0.38 to 0.44. That is a red flag that a large fraction of the benchmark may be solvable from structured mask-derived statistics, rather than from the kind of image-grounded reasoning the paper claims. This matters a lot scientifically, because otherwise the work risks measuring alignment between benchmark generation and tool outputs, rather than general 3D understanding. The paper needs explicit tool-only or mask-statistics baselines to show what portion of the gain truly comes from the agent.

3：Equations (4) to (6) on Page 5 define the T1S-Loop at a high level, but the actual routing policy is vague. The paper says the router decides whether more evidence is needed and which tool to apply, but it does not concretely specify the prompt, decision criteria, or slice/ROI selection policy in the main paper. There is also notation drift: in Equation (4), (r_t) is introduced as a rationale, but in the following paragraph (r_t) is used again as a slice or ROI to be selected, which is confusing. Similarly, the text defines (\mathcal{A}_t) but Equation (4) writes (a_t). These are not cosmetic issues only, because the paper’s contribution is largely procedural. If the procedure is underspecified, the claimed method becomes hard to reproduce and hard to analyze.

4：At a high level, the system combines existing 3D visual tools, a 2D MLLM, a memory bank, and an iterative reasoning loop. OAMI is essentially structured organ-statistics extraction; CFLT repurposes CT-CLIP local similarities plus organ filtering; T1S-Loop is an iterative tool-use loop with memory updates. This is a sensible engineering composition, but the paper does not convincingly isolate what is technically new beyond the packaging and task framing. Figure 2, Page 4, makes the architecture look polished, but the main components themselves are fairly standard ideas in tool-augmented agent systems. For a paper making broad claims about a unified 3D analysis paradigm, stronger evidence of methodological distinctiveness would help.

**Limitations:**

yes

**Strengths And Weaknesses:**

Strength:
The paper tackles a real and important problem, namely how to bridge low-level 3D perception and higher-level clinical reasoning without building yet another heavily fine-tuned 3D foundation model. The overall decomposition is sensible: OAMI provides global structured cues, CFLT narrows attention to lesion-relevant regions, and T1S-Loop adds targeted verification. In that sense, the work has a coherent systems design rather than a bag of disconnected tricks.

The presentation of the pipeline is mostly intuitive. Figure 2 is particularly helpful in showing how OAMI, CFLT, and T1S-Loop interact through a shared memory, and Figure 3 gives concrete workflows for measurement, recognition, and medical reasoning tasks. These figures make the agentic story much easier to follow than the equations alone.

Empirically, the results are strong on paper. In Table 2, the overall average on DeepTumorVQA improves from 0.42 for the best listed baseline to 0.66 for 3DMedAgent with GPT-5. In Table 3, the total average on DeepChestVQA increases from 0.45 for the best listed baseline to 0.57. The ablation in Table 4 also suggests that all three proposed components contribute.

The benchmark contribution is potentially useful. Chest CT is underrepresented relative to abdominal 3D VQA, and the taxonomy in Table 1 covers recognition, visual reasoning, and medical reasoning rather than a single narrow task.

Weakness:
1:lack of novelty. Much of the system is an orchestration of existing ingredients: a 3D segmentation model, a pretrained CT-text model, prompting, memory summarization, and tool routing. At a high level, the system combines existing 3D visual tools, a 2D MLLM, a memory bank, and an iterative reasoning loop. OAMI is essentially structured organ-statistics extraction; CFLT repurposes CT-CLIP local similarities plus organ filtering; T1S-Loop is an iterative tool-use loop with memory updates. This is a sensible engineering composition, but the paper does not convincingly isolate what is technically new beyond the packaging and task framing. Figure 2, Page 4, makes the architecture look polished, but the main components themselves are fairly standard ideas in tool-augmented agent systems. For a paper making broad claims about a unified 3D analysis paradigm, stronger evidence of methodological distinctiveness would help.

2. In the main paper, DeepChestVQA is built from organ and lesion masks, with “relevant indicators computed from the masks to derive answers” for medical reasoning questions, see Section 3, Page 3. Meanwhile, OAMI computes organ size, mean HU, and z-range directly from segmentation masks, and CFLT localizes lesion ROIs using CT-CLIP heatmaps constrained by organ memory, see Sections 4.1 to 4.2, Pages 4 to 5. Table 4 on Page 8 makes this especially visible: adding OAMI alone, with text-only MLLM input, boosts DeepTumorVQA medical reasoning from 0.43 to 0.62 and DeepChestVQA visual reasoning from 0.38 to 0.44. That is a red flag that a large fraction of the benchmark may be solvable from structured mask-derived statistics, rather than from the kind of image-grounded reasoning the paper claims. This matters a lot scientifically, because otherwise the work risks measuring alignment between benchmark generation and tool outputs, rather than general 3D understanding. The paper needs explicit tool-only or mask-statistics baselines to show what portion of the gain truly comes from the agent.

3. Equations (4) to (6) on Page 5 define the T1S-Loop at a high level, but the actual routing policy is vague. The paper says the router decides whether more evidence is needed and which tool to apply, but it does not concretely specify the prompt, decision criteria, or slice/ROI selection policy in the main paper. There is also notation drift: in Equation (4), (r_t) is introduced as a rationale, but in the following paragraph (r_t) is used again as a slice or ROI to be selected, which is confusing. Similarly, the text defines (\mathcal{A}_t) but Equation (4) writes (a_t). These are not cosmetic issues only, because the paper’s contribution is largely procedural. If the procedure is underspecified, the claimed method becomes hard to reproduce and hard to analyze.

4: Figure 6 on Page 8 is presented positively, but the top-1 agreement between the method and radiologists looks only around 0.3 to 0.35, while radiologist-radiologist top-1 agreement is much higher, roughly 0.88. The top-3 agreement is indeed much better, but that means the method is good at proposing a short list, not necessarily at selecting the single best slice. Likewise, Figure 7 on Page 8 shows that accuracy drops as iteration turns increase. The authors interpret this as harder cases being routed to more turns, which is plausible, but the figure does not by itself establish that the loop is robust or efficient. These are not fatal flaws, but the current text leans a bit too hard into the optimistic interpretation.

5: In the main paper, DeepChestVQA is built from organ and lesion masks, with “relevant indicators computed from the masks to derive answers” for medical reasoning questions, see Section 3, Page 3. Meanwhile, OAMI computes organ size, mean HU, and z-range directly from segmentation masks, and CFLT localizes lesion ROIs using CT-CLIP heatmaps constrained by organ memory, see Sections 4.1 to 4.2, Pages 4 to 5. Table 4 on Page 8 makes this especially visible: adding OAMI alone, with text-only MLLM input, boosts DeepTumorVQA medical reasoning from 0.43 to 0.62 and DeepChestVQA visual reasoning from 0.38 to 0.44. That is a red flag that a large fraction of the benchmark may be solvable from structured mask-derived statistics, rather than from the kind of image-grounded reasoning the paper claims. This matters a lot scientifically, because otherwise the work risks measuring alignment between benchmark generation and tool outputs, rather than general 3D understanding. The paper needs explicit tool-only or mask-statistics baselines to show what portion of the gain truly comes from the agent.

---

> ### Author Rebuttal · Authors · 2026-03-30
>
> Thanks for the thorough evaluation. We appreciate the recognition of our coherent systems design, strong results, and benchmark contribution. We address the concerns below.
>
> **W1/Q4:**
> We respectfully argue that our contribution is a **paradigm-level innovation**. Much like ReAct (Yao et al., 2022) contributed a reasoning-acting paradigm rather than new LLMs or tools, ours lies in enabling a 2D MLLM to **actively explore 3D imaging at adaptive granularity** conditioned on the query:
>
> **1. Design over tool dependence.** Replacing VISTA3D with TotalSegmentator in OAMI:
>
> | Setting               | Mea. | Rec. | Visual Rea. | Medical Rea. | Overall |
> | --------------------- | ---- | ---- | ----------- | ------------ | ------- |
> | GPT-5 (baseline)      | 0.33 | 0.49 | 0.39        | 0.43         | 0.41    |
> | 3DMedAgent (VISTA3D)  | 0.63 | 0.76 | 0.56        | 0.70         | 0.66    |
> | 3DMedAgent (TotalSeg) | 0.58 | 0.76 | 0.55        | 0.67         | 0.64    |
>
> Reasoning tasks remain stable across tools. The framework is also **backbone-agnostic** (*Tables 2 and 3*), working with both GPT-5 and Qwen3-VL with consistent gains.
>
> **2. Non-trivial technical design.** CT-CLIP is a **volume-level** model not designed for fine-grained. CFLT repurposes it into a **region-level** localization tool via dense local features and organ-aware heatmap ranking (*Section 4.2*). This is a methodological adaptation, not simple integration.
>
> **3. Agent-driven, multi-granularity exploration.** Feeding raw tool outputs without agentic interaction yields substantial degradation on reasoning tasks (*see Q1 below*), confirming the exploration paradigm derives the most gains.
>
> **Q1:**
> We construct the mentioned tool-aware baseline, feeding the raw tool outputs to GPT-5 for **single-round VQA** without agentic interaction:
>
> | Setting                  | Mea. | Rec. | Visual Rea. | Medical Rea. | Overall |
> | ------------------------ | ---- | ---- | ----------- | ------------ | ------- |
> | GPT-5 (baseline)         | 0.33 | 0.49 | 0.39        | 0.43         | 0.41    |
> | GPT-5 + raw tool outputs | 0.55 | 0.73 | 0.45        | 0.58         | 0.58    |
> | 3DMedAgent (full)        | 0.63 | 0.76 | 0.56        | 0.70         | 0.66    |
>
> Tool outputs get a significant gain, but **substantial gap remains on Visual Reasoning (+0.11) and Medical Reasoning (+0.12)**. This shows the performance gain stems from the framework instead of tools alone.
>
> **W2/W5/Q2:**
>
> **Benchmark construction.** DeepChestVQA is not trivially solvable from mask statistics. Lesion annotations come from ReXGroundingCT (Baharoon et al., 2025), provided by experts, **not** generated by segmentation models. 2) We also excluded **Measurement** questions from DeepChestVQA since they can be derived from masks. The remaining types require deeper analysis and evidence integration beyond single mask statistics.
>
> **Agent reasoning is essential.** OAMI deliberately does not perform lesion segmentation (*Section 4.1*). Lesion-related reasoning requires CFLT, where the agent infers target organs, retrieves prior memory, and scores ROIs as candidates. Raw tool outputs are not directly usable by 2D MLLM: segmentation masks are 3D volumes and CT-CLIP scores provide no spatial context.
>
> **Experimental validation.** The tool-only baseline in *Q1* shows a significant overall gap (0.58 vs. 0.66), especially on reasoning tasks, confirming that **gains come from progressive reasoning, not mask statistics alone**.
>
> **W3/Q3:**
>
> Full prompts and workflow are in our anonymized code, and we will also reformulate part of *Section 4.3* into pseudocode:
> ```
> Input: query q, memory M, max iterations T_max
> Output: final answer y_hat
>
> for t = 0 to T_max do
>     (r_t, y_t, E_t, a_t) <- Agent(q, M)        // reasoning, answer, evidence, assumptions
>     (b_t, tau_t) <- Router(r_t, y_t, E_t, a_t) // continue? which tool?
>     if b_t = 0: return y_t                     // evidence sufficient
>     s_t<- SelectSlice(M, a_t)                  // unvisited slice/ROI
>     M <- M ∪ {ApplyTool(tau_t, s_t), r_t, E_t, a_t}
> end for
> return y_{T_max}
> ```
> Notation fixes: $r_t$ (slice/ROI) will be renamed to $s_t$; $\mathcal{A}_t$ vs. $a_t$ is a typo and we will unify throughout. We have reviewed the full manuscript for similar issues.
>
> **W4:**
> We agree a more balanced presentation is warranted.
>
> **Figure 6:** 1) Clinically, radiologists examine multiple slices rather than a single one; top-1 selection is inherently subjective. 2) CFLT is designed to **produce a candidate shortlist** for T1S-Loop, not finalize a single slice. The high top-3 agreement confirms CFLT fulfills this goal. We will contextualize the top-1 gap more carefully.
>
> **Figure 7:** We clarify that the x-axis groups **different subsets** of questions by the iteration turns assigned (see response to Review vNBv Q4), showing T1S-Loop benefits all difficulty levels. We agree previous discussion was too optimistic and will revise analysis for a balanced presentation.

---

> > ### Author Rebuttal · Reviewer_B7wS · 2026-04-03
> >
> > The rebuttal to the weakness still does not convince me too much. I thus keep my rating

---

> > > ### Author Response · Authors · 2026-04-03
> > >
> > > Thank you very much for your feedback. We really appreciate your time and consideration.
> > >
> > > To further improve the paper and better meet your expectations, we would sincerely appreciate it if you could share any remaining issues or factors that are currently preventing you from raising the score. We are more than happy to provide additional clarifications, conduct further analysis, or revise the paper accordingly. Your guidance would be extremely valuable in helping us strengthen the work.

---

### Official Review · Reviewer_692C · 2026-03-12

**Soundness:** 3
**Presentation:** 4
**Significance:** 3
**Originality:** 4
**Overall Recommendation:** 4
**Confidence:** 3

**Summary:**

The paper proposes 3DMedAgent, an agent-based framework designed to connect low-level perception and high-level reasoning for 3D CT analysis using multimodal large language models. The system enables a 2D MLLM to analyze 3D data by orchestrating external perception tools, converting volumetric information into informative slices, and storing intermediate outputs as structured evidence in a shared memory for multi-step reasoning. The framework includes organ-aware memory initialization to provide global anatomical context, a coarse-to-fine lesion targeting strategy to identify candidate regions of interest, and an iterative slice-based reasoning loop that verifies and refines evidence. Experiments on DeepTumorVQA and the newly introduced DeepChestVQA benchmark demonstrate that the approach substantially improves performance compared with general MLLMs, medical vision-language models, and specialized 3D models.

**Compliance With Llm Reviewing Policy:**

Affirmed.

**Final Justification:**

After reading the rebuttal comments, I decided to keep my score.

**Key Questions For Authors:**

1. Could the authors clarify how the proposed system differs from other recent agent-based multimodal frameworks beyond system-level orchestration?
2. Some implementation details regarding the interaction between the agent and external tools are not fully described. Could the authors elaborate on the prompting strategies, tool invocation protocol, and memory update mechanisms to ensure that the framework can be reliably reproduced?

**Limitations:**

yes

**Strengths And Weaknesses:**

The submission addresses a problem in enabling multimodal language models to analyze volumetric medical imaging and proposes a coherent agent-based framework that integrates perception tools, structured memory, and iterative reasoning. The methodology is technically reasonable and supported by empirical evaluation across multiple benchmarks, including the DeepChestVQA dataset, and the experiments include comparisons with several general, medical and 3D vision-language models as well as ablation studies demonstrating the contributions of key components. The framework is clearly structured, and the modular design provides an interpretable pipeline that mirrors clinical reasoning workflows, suggesting potential for future research on tool-augmented multimodal systems for complex medical imaging tasks.

However, the experimental analysis focuses largely on benchmark accuracy improvements and provides limited investigation into the internal reasoning behavior of the agent, making it difficult to determine whether performance gains arise from improved reasoning or from more effective evidence retrieval. In addition, some implementation details regarding the interaction between the agent and external tools, as well as prompting strategies used during inference, are not fully specified, which may limit the reproducibility for future work.

---

> ### Author Rebuttal · Authors · 2026-03-30
>
> Sincerely thanks for the positive recognition of our framework and the DeepChestVQA benchmark. We address each concern below.
>
> **W1: Internal reasoning behavior:**
>
> We agree deeper analysis beyond accuracy would strengthen the paper. Our results indicate that **agentic reasoning is the primary contribution of complex tasks**. Directly feeding raw tool outputs to GPT-5 for single-round VQA yields 0.45 on Visual Reasoning and 0.58 on Medical Reasoning, whereas the full 3DMedAgent achieves 0.56 and 0.70:
>
> | Setting                  | Mea. | Rec. | Visual Rea. | Medical Rea. | Overall |
> | ------------------------ | ---- | ---- | ----------- | ------------ | ------- |
> | GPT-5 (baseline)         | 0.33 | 0.49 | 0.39        | 0.43         | 0.41    |
> | GPT-5 + raw tool outputs | 0.55 | 0.73 | 0.45        | 0.58         | 0.58    |
> | 3DMedAgent (full)        | 0.63 | 0.76 | 0.56        | 0.70         | 0.66    |
>
> This confirms that **progressive exploration at hierarchical granularity, not evidence retrieval alone, accounts for the performance gains on reasoning tasks**. The current submission also includes analyses of internal behavior: slice-selection agreement with radiologists (*Figure 6*), iteration analysis (*Figure 7*), and step-by-step case studies including failure cases (*Figure 3 and Appendix C.2*). We will add more detailed qualitative analysis of reasoning steps and failure patterns in the revision.
>
> **W2 & Q2: Reproducibility and implementation details.**
>
> We address reproducibility at three levels:
>
> **1. Codebase.** Full prompts, tool invocation protocols, and memory update logic are in our anonymized codebase (*linked in the abstract*). We will improve code quality and add annotations.
>
> **2. Paper revision.** We will reformulate *Section 4.3* into pseudocode specifying 3DMedAgent’s routine criteria (*see response to Reviewer B7wS W3*), with additional examples in the *Appendix*.
>
> **3. Implementation specifics:**
>
> - **Prompting strategy:** At each step, the MLLM agent receives structured input with fields for the current query and shared memory (organ/lesion/iteration memory), and available actions. A JSON output template defines required keys (e.g., `evidence` field should contain specific memory entries supporting the current decision, and the `action` field specifies the next step). This ensures inputs and outputs remain structured and parseable throughout reasoning.
> - **Tool invocation protocol:** Available actions and parameters are enumerated as a list in the prompt, each mapped to an executable function. When the MLLM output conforms to the JSON format, the corresponding function is invoked directly. This bridges textual reasoning and programmatic tool execution.
> - **Memory update mechanism:** After each tool call or slice inspection, the output is summarized into a compact textual entry (e.g., "Kidney: size 307 cm³, mean HU 148.9, z-range [67,82]") and appended to memory. During T1S-Loop, each iteration also records rationale, answer, evidence, and assumptions (*Eq. 4*). Visited slices are marked to prevent redundant inspection.
>
> **Q1: Difference from other agent-based frameworks**
>
> As the reviewer noted, our framework mirrors clinical reasoning workflows by design, distinguishing it from general-purpose agent architectures. Details are below:
>
> **1. Agent-driven, multi-granularity exploration.** Previous agent frameworks (e.g., MMedAgent, MedAgent-Pro) use tools reactively. 3DMedAgent follows an evidence-based, 3D domain-specific paradigm: the MLLM actively decides what to explore and at what granularity (OAMI at the organ level, CFLT at the lesion level, T1S-Loop at the ROI level). This coarse-to-fine progression covers the full spectrum from perception to understanding without task-specific training.
>
> **2. Tools as perception instruments, not static integration.** Raw tool outputs are not directly usable by the MLLM: a segmentation mask is a 3D volume a 2D MLLM cannot process, and a CT-CLIP score provides no spatial context. The agent bridges this through structured interaction: inferring what is needed, invoking tools, and distilling outputs into compact evidence. The tool-only comparison in *W1* confirms this further.
>
> **3. Framework design over tool dependence.** Replacing VISTA3D with TotalSegmentator preserves reasoning performance:
>
> | Setting               | Mea. | Rec. | Visual Rea. | Medical Rea. | Overall |
> | --------------------- | ---- | ---- | ----------- | ------------ | ------- |
> | GPT-5 (baseline)      | 0.33 | 0.49 | 0.39        | 0.43         | 0.41    |
> | 3DMedAgent (VISTA3D)  | 0.63 | 0.76 | 0.56        | 0.70         | 0.66    |
> | 3DMedAgent (TotalSeg) | 0.58 | 0.76 | 0.55        | 0.67         | 0.64    |
>
> These results demonstrate that the **exploration paradigm itself**, rather than specific tools or simple orchestration, is the primary contributor, and our framework is not limited by specific tool choice.

---

> > ### Author Rebuttal · Reviewer_692C · 2026-04-03
> >
> > I appreciate the authors’ detailed response and clarification. My questions are addressed.

---

> > > ### Author Response · Authors · 2026-04-04
> > >
> > > Thank you very much for your positive evaluation. Your feedback has helped improve the quality and clarity of our manuscript.

---

### Official Review · Reviewer_kuT7 · 2026-03-13

**Soundness:** 3
**Presentation:** 3
**Significance:** 3
**Originality:** 2
**Overall Recommendation:** 4
**Confidence:** 4

**Summary:**

This paper proposes 3DMedAgent, an agent-based framework that enables 2D multimodal large language models (MLLMs) to perform unified 3D CT analysis without requiring 3D-specific fine-tuning. The system decomposes volumetric analysis into sequential perception and reasoning steps by invoking specialized visual tools and storing intermediate results in a textual memory for evidence-based reasoning. Key components include Organ-Aware Memory Initialization (OAMI), Coarse-to-Fine Lesion Targeting (CFLT), and a Think-with-1-Slice Loop (T1S-Loop) that iteratively verifies visual evidence. The authors also introduce the DeepChestVQA benchmark for evaluating 3D medical understanding, and experiments across more than 40 tasks show the effectiveness of the proposed approach.

**Compliance With Llm Reviewing Policy:**

Affirmed.

**Final Justification:**

Thank you for the clarification. My previous concerns have been addressed, and I have increased my score accordingly.

**Key Questions For Authors:**

Please see the Weaknesses.

**Limitations:**

yes

**Strengths And Weaknesses:**

*Strengths
1. The paper presents extensive experiments across multiple datasets and tasks, including the proposed benchmark, with comparisons to a range of general, medical, and 3D-specific MLLMs, providing a thorough empirical validation of the approach.
2. Developing agents that can effectively analyze 3D medical imaging data is an important direction with clear clinical relevance, and the proposed framework could have potential value for assisting medical image analysis and decision support.

*Weaknesses
1. The paper argues that existing methods compress large 3D volumes into limited tokens for 2D MLLM backbones, which may discard spatial context. However, the proposed framework also converts 3D information into textual representations by CT-aware tools (e.g., VISTA3D and CT-CLIP) and stores them in a textual memory (prompt). It is therefore unclear why this text would preserve 3D information more effectively. Is the improvement mainly due to the use of specialized CT-aware tools for extracting evidence? If so, it would be helpful to clarify how this differs fundamentally from existing methods such as Merlin or CT-CHAT, which integrate 3D vision encoders into existing 2D MLLM frameworks. And why the proposed method should lead to better performance, or what the underlying principle behind the improvement is.
2. The technical novelty of the textual memory mechanism is limited. The framework maintains a textual memory bank that stores intermediate evidence for reasoning. However, similar ideas have already been explored in several agent-based LLM systems, where intermediate results are stored as structured memory and reused during multi-step reasoning or dialogue (e.g., [1,2]). Since these works also treat memory as explicit evidence for reasoning, it would be helpful to clarify what is technically novel in the proposed memory design or how it differs from existing memory-based agent frameworks.
3. The contribution appears largely system-level and may be primarily engineering. From the current version, the main contribution seems to be the construction of the 3DMedAgent framework, which orchestrates CT-aware tools and converts their outputs into textual evidence that can be used by a 2D MLLM during reasoning. If my understanding is correct, the paper may be primarily a system integration effort rather than introducing fundamentally new ideas.

[1] Memory-R1: Enhancing Large Language Model Agents to Manage and Utilize Memories via Reinforcement Learning. 2025.

[2] In Prospect and Retrospect: Reflective Memory Management for Long-term Personalized Dialogue Agents. ACL 2025.

---

> ### Author Rebuttal · Authors · 2026-03-30
>
> Thanks for the thoughtful questions. We first provide a high-level perspective: the core innovation of 3DMedAgent is **enabling a 2D MLLM to actively explore 3D imaging on demand, progressively, and at hierarchical granularity**, covering the full spectrum from perception to clinical understanding. **The MLLM decides the exploration action, while tools serve as its perception instruments.** Much like ReAct (Yao et al., 2022) contributed a reasoning-acting paradigm rather than new LLMs or tools, our contribution lies in this coarse-to-fine agentic exploration paradigm tailored for 3D medical imaging.
>
> **W1:**
> The key distinction between 3DMedAgent and end-to-end 3D MLLMs is **not** text vs. visual tokens, but **active, query-specific exploration vs. one-hop compression**. 3D MLLMs compress the entire volume into fixed tokens for all tasks, inevitably discarding fine-grained details. 3DMedAgent takes a different paradigm: the MLLM actively decides what to explore at what granularity based on the task, and conducts 3D perception through tools (OAMI for organ-level, CFLT for lesion-level, and T1S-Loop for ROI-level). The textual memory serves as a **task-conditioned evidence storage** that accumulates multi-granularity evidence, rather than replacing the 3D visual representation.
>
> To verify this, we compare against feeding raw tool outputs to GPT-5 for single-round VQA:
>
> | Setting                  | Mea. | Rec. | Visual Rea. | Medical Rea. | Overall |
> | ------------------------ | ---- | ---- | ----------- | ------------ | ------- |
> | GPT-5 (baseline)         | 0.33 | 0.49 | 0.39        | 0.43         | 0.41    |
> | GPT-5 + raw tool outputs | 0.55 | 0.73 | 0.45        | 0.58         | 0.58    |
> | 3DMedAgent (full)        | 0.63 | 0.76 | 0.56        | 0.70         | 0.66    |
>
> **A substantial gap remains on Visual Reasoning (0.45 vs. 0.56) and Medical Reasoning (0.58 vs. 0.70)**, confirming that the agent's progressive exploration drives the gains on reasoning tasks, not tool outputs alone.
>
> 3D MLLMs such as Merlin and CT-CHAT both integrate 3D vision encoders into 2D MLLMs (*Section 2.1*), sharing **the bottleneck of one-pass volume compression**. We further evaluate CT-CHAT-13B on DeepTumorVQA:
>
> | Setting    | Mea. | Rec. | Visual Rea. | Medical Rea. |
> | ---------- | ---- | ---- | ----------- | ------------ |
> | CT-CHAT    | 0.21 | 0.51 | 0.31        | 0.36         |
> | 3DMedAgent | 0.63 | 0.76 | 0.56        | 0.70         |
>
> CT-CHAT underperforms substantially on Measurement and Medical Reasoning, where fine-grained perception and evidence integration are essential, confirming that fixed compression lacks the adaptive granularity for diverse clinical tasks.
>
> **W2:**
> The cited works address different settings. Memory-R1 and RMM manage memories **across sessions or interactions** for cross-session memory reuse. Our memory operates within a **single task**, storing **perceptual evidence progressively** gathered from 3D imaging tools (organ statistics from OAMI, lesion candidates from CFLT, and ROI observations from T1S-Loop). The memory grows adaptively as the agent explores at hierarchical granularity, enabling query-conditioned cue acquisition and aggregation.
>
> The novelty of our memory design lies not in the management mechanism (update/retrieve), but in **its content and motivation**:  it is tailored for progressive 3D medical exploration from coarse to fine.
> The ablation supports this in *Table 4*, where each exploration stage brings consistent gains, confirming that cross-step evidence integration is essential. We will also discuss the mentioned works in the revised **Related Work** section.
>
> **W3:**
> We highlight additional supporting points beyond **paradigm-level innovation** discussed above:
>
> **Tool Substitutability.** Replacing VISTA3D with TotalSegmentator (another segmentation model)  in OAMI:
>
> | Setting               | Mea. | Rec. | Visual Rea. | Medical Rea. | Overall |
> | --------------------- | ---- | ---- | ----------- | ------------ | ------- |
> | GPT-5 (baseline)      | 0.33 | 0.49 | 0.39        | 0.43         | 0.41    |
> | 3DMedAgent (VISTA3D)  | 0.63 | 0.76 | 0.56        | 0.70         | 0.66    |
> | 3DMedAgent (TotalSeg) | 0.58 | 0.76 | 0.55        | 0.67         | 0.64    |
>
> Measurement drops due to segmentation quality differences, but **reasoning tasks remain stable**, confirming that the exploration design, not specific tools, drives the gain. The framework is also **backbone-agnostic**: as shown in *Tables 2 and 3*, it consistently improves the baselines for both GPT-5 and Qwen3-VL.
>
> **Technical design.** CT-CLIP is a volume-level contrastive model; CFLT repurposes it into a fine-grained localization tool (*Section 4.2*). This is a methodological contribution, not standard tool integration.
>
> **Additional contributions.** We introduce the **DeepChestVQA benchmark** across 17 capability dimensions for thoracic CT, filling a gap in 3D medical evaluation.

---

> > ### Author Rebuttal · Reviewer_kuT7 · 2026-04-04
> >
> > My previous concerns have been addressed, and I have increased my score accordingly.

---

> > > ### Author Response · Authors · 2026-04-04
> > >
> > > We are glad that our responses have addressed your concerns. Thank you for the constructive suggestions and postive feedback, and we will incorporate the revisions into the paper.

---

### Decision · Program_Chairs · 2026-04-30

**Decision:**

Accept (regular)

**Comment:**

This submission tackles an important problem and presents a practical, well-executed framework for extending 2D MLLMs to 3D CT analysis through tool augmentation and multi-step reasoning. Reviewers generally agreed on the importance of the setting, the strength of the empirical results, and the value of the new DeepChestVQA benchmark. The main concerns were about the degree of methodological novelty, the extent to which the gains arise from the agentic reasoning versus the strength of the external 3D tools, and some missing implementation details affecting reproducibility. The rebuttal addressed several of these concerns for most reviewers, although one reviewer remained unconvinced on novelty and evaluation fairness.

 Overall, I view the contribution as primarily system-level but still meaningful and above the bar, given the strong experimental evidence, practical relevance, and benchmark contribution.

In addition, the negative reviewer did not provide the detailed concerns in the further disucssion and final discussion. I suggest the authors to refine all draft by merging the rebuttal experiments.